# Multiple conformations facilitate PilT function in the type IV pilus

Matthew McCallum[1,2], Samir Benlekbir[2], Sheryl Nguyen[2], Stephanie Tammam[2], John L. Rubinstein [1,2,3]*, Lori L. Burrows [4]* & P. Lynne Howell [1,2]*

Type IV pilus-like systems are protein complexes that polymerize pilin fibres. They are critical for virulence in many bacterial pathogens. Pilin polymerization and depolymerization are powered by motor ATPases of the PilT/VirB11-like family. This family is thought to operate with $C_2$ symmetry; however, most of these ATPases crystallize with either $C_3$ or $C_6$ symmetric conformations. The relevance of these conformations is unclear. Here, we determine the X-ray structures of PilT in four unique conformations and use these structures to classify the conformation of available PilT/VirB11-like family member structures. Single particle electron cryomicroscopy (cryoEM) structures of PilT reveal condition-dependent preferences for $C_2$, $C_3$, and $C_6$ conformations. The physiologic importance of these conformations is validated by coevolution analysis and functional studies of point mutants, identifying a rare gain-of-function mutation that favours the $C_2$ conformation. With these data, we propose a comprehensive model of PilT function with broad implications for PilT/VirB11-like family members.

---

[1] Department of Biochemistry, University of Toronto, Toronto, ON M5S 1A8, Canada. [2] Program in Molecular Structure & Function, Peter Gilgan Centre for Research and Learning, The Hospital for Sick Children, Toronto, ON M5G 0A4, Canada. [3] Department of Medical Biophysics, University of Toronto, Toronto, ON M5G 1l7, Canada. [4] Department of Biochemistry and Biomedical Sciences and the Michael G. DeGroote Institute for Infectious Disease Research, McMaster University, Hamilton, ON L8S 4K1, Canada. *email: john.rubinstein@utoronto.ca; burrowl@mcmaster.ca; howell@sickkids.ca

Type IV pilus-like (T4P-like) systems are distributed across all phyla of prokaryotic life[1,2]. T4P-like systems include the type IVa pilus (T4aP), type II secretion (T2S) system, type IVb pilus (T4bP), Tad/Flp pilus (T4cP), Com pilus, and archaellum — with the latter found exclusively in Archaea. These systems enable attachment, biofilm formation, phage adsorption, surface-associated or swimming motility, natural competence, and folded protein secretion in bacteria and Archaea[3–5], and thus are of vital medical and industrial importance. As many of these systems are critical for virulence in bacterial pathogens[1,6–8]; conserved components could have value as therapeutic targets. Despite the importance of T4P-like systems, basic questions — including how the pilus is assembled and disassembled — remain open.

All systems have at least three conserved and essential elements: a pilus polymer of subunits termed pilins, a pre-pilin peptidase, and a motor[9]. The pre-pilin peptidase cleaves the N-terminal leader peptides of pilin subunits at the inner face of the cytoplasmic membrane, leaving the mature pilins embedded in the membrane[10]. Polymerization requires extraction of pilins from the membrane by the cytoplasmic motor by using energy generated from ATP hydrolysis[11–13]. The motor is made up of two well-conserved components: a cytoplasmic ring-like hexameric PilT/VirB11-like ATPase and a PilC-like inner-membrane platform protein[9]. As mature pilins cannot interact directly with the cytoplasmic ATPases, their polymerization requires that both the ATPase and pilins interact with the PilC-like platform protein[9]. Cryo-electron tomography (cryoET) studies of the T4aP, T4bP, archaellum, and T2S systems are consistent with localization of the PilC-like protein in the pore of the hexameric ATPase, connecting it to the pilus on the exterior of the inner membrane[14–17]. The ATPase connects to stator-like components, suggesting that a fixed ATPase moves the PilC-like protein[15]. Thus, PilT/VirB11-like family members are thought to power pilin polymerization by rotating the PilC-like protein to extract pilins from the membrane and inserting them into the base of the growing pilus polymer[15,18].

Detailed structural analysis of PilT/VirB11-like ATPases has advanced our understanding of how the T4P-like motor could insert pilins into the base of a helical pilus. They form hexamers that can be represented as six rigid subunits, termed packing units, held together by flexible linkers[18] (Supplementary Fig. 1). Adjacent packing units adopt one of two conformations: open (O) or closed (C)[18]. PilB is the ATPase that powers pilin polymerization in the T4aP. All PilB motor structures determined to date are similar in overall conformation: they exhibit $C_2$ symmetry with a CCOCCO pattern of open- and closed interfaces around the hexamer. In this conformation, the pore of PilB is elongated. Using the heterogeneous distribution of nucleotides in ADP-bound and ADP/ATP-analog-bound PilB crystal structures, we deduced that ATP binding and hydrolysis in the CCOCCO PilB structure would propagate conformational changes leading to a clockwise rotation of the elongated pore[18]. PilC, bound in the pore, would thus be turned clockwise in 60° increments, while accompanying conformational changes in the PilB subunits would displace PilC out of the plane of the inner membrane, toward the periplasm[18]. If a pilin is inserted at each clockwise increment, these motions would build a one-start, right-handed helical pilus[18], consistent with cryoEM structures[19,20].

T4aP polymers can be rapidly depolymerized at the base, resulting in fiber retraction. PilT is the PilT/VirB11-like ATPase that powers T4aP depolymerisation[21]. We applied the same analysis used to deduce the movements of PilB to the $C_2$ symmetric structure of PilT from *Aquifex aeolicus* (PilT[Aa], PDB 2GSZ[22])[18]. We found that this protein had an OOCOOC pattern of interfaces, which would give the impression of

counterclockwise rotation of the elongated pore and potential downward movement of PilC[18]. Thus, we proposed that PilT may act like PilB in reverse, consistent with powering pilus depolymerization[18]. This analysis highlighted the importance of clarifying the symmetry and pattern of open- and closed interfaces in PilT/VirB11-like family members when interpreting their structures and defining mechanisms.

In contrast to PilB structures, which exhibit only $C_2$ symmetry, PilT has been crystallized in a variety of conformations with $C_6$ symmetry[22–24]. Other PilT/VirB11-like family members have crystallized in conformations with $C_2$, $C_3$, and $C_6$ symmetries[12,25,26]. Multiple potential crystallographic conformations of PilT and PilT/VirB11-like family members suggest that the OOCOOC conformation may not represent the active PilT retraction motor[22,27]. Further, the OOCOOC PilT structure was determined by using a homolog from Aquificae[22] and may not reflect a conformation typical of PilT from Proteobacteria, where most of the phenotypic analyses of the T4aP have been conducted. The specific conformation adopted by the PilT motor, and the details of retraction remain to be clarified.

Here, we crystallize PilT in four unique conformations, including the highest resolution structure to date of a hexameric PilT/VirB11-like family member. These structures allow the identification of conserved open- and closed-interface contact points that are used to differentiate the conformations of available PilT/VirB11-like structures into six unique classes. To examine the conformations adopted by PilT and PilB in a noncrystalline state, we determine their structures by using cryoEM. Those structures of PilT reveal a clear preference for $C_2$ and $C_3$ conformations in the absence of nucleotide or with ADP, and the $C_6$ conformation in the presence of ATP. These structures are validated by coevolution analysis and functional analysis of point mutants. A gain-of-function mutation with increased in vivo activity is identified, and cryoEM analysis reveals its preference for the $C_2$ conformation. From these data, we propose a comprehensive model of PilT function with broad implications for all PilT/VirB11-like family members.

## Results

**PilT crystallizes in $C_3$ and pseudo-$C_3$ symmetric conformations.** To gauge the reproducibility of previously crystallized PilT conformations and to see if additional conformations could be identified, PilT4 from *Geobacter metallireducens* (PilT[Gm]) was crystallized. PilT[Gm] was selected because we previously crystallized PilB from *G. metallireducens* to derive models for PilB-mediated extension[18]. There are four PilT orthologs in *Geobacter*, and PilT4 is the primary retraction ATPase in *Geobacter sulfurreducens*[28]. In the absence of added nucleotide, PilT[Gm] crystallized only after reductive methylation. These crystals diffracted to 3.3 Å, and the structure, determined by molecular replacement, revealed a $C_3$ symmetric hexamer in the asymmetric unit (Fig. 1a). No nucleotide was found in this structure, although density consistent with sulfate, present in the crystallization conditions, was observed in the nucleotide-binding site (Fig. 1b). By comparison with the interfaces in PilB[Gm], we categorized the interfaces between the packing units of PilT[Gm] as alternating between open and closed. This OCOCOC conformation has not previously been reported for PilT.

We hypothesized that this conformation in the absence of a nucleotide could have resulted from the methylation process, rather than representing a physiologically relevant PilT conformation. Therefore, to crystallize PilT[Gm] in the absence of added nucleotide without reductive methylation, we extensively optimized the buffer (see the "Methods" section). The resulting

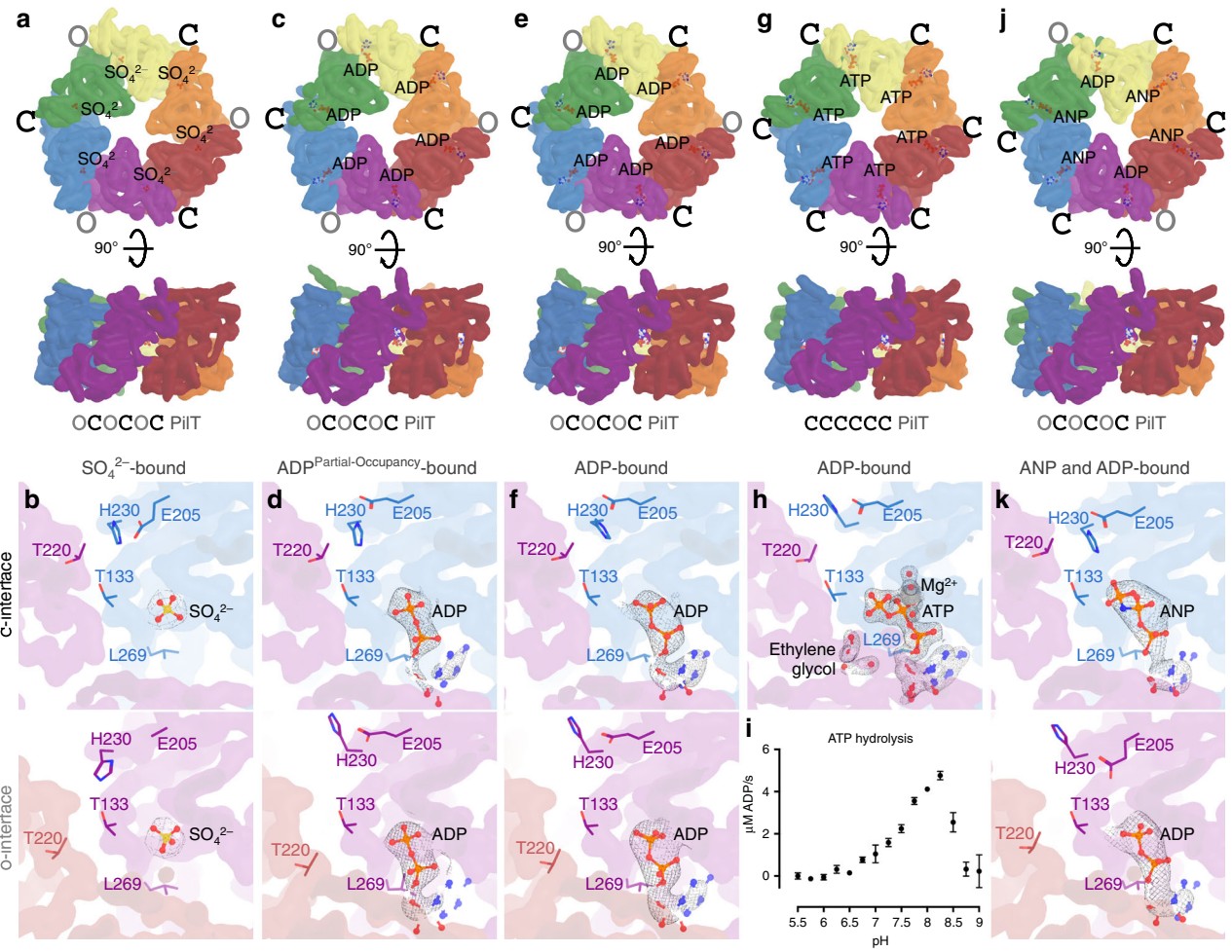

**Fig. 1** PilT from *G. metallireducens* (PilT$^{Gm}$) crystallizes in multiple conformations. Individual packing units (N2D$^n$ and CTD$^{n+1}$) are uniquely colored. A smoothed transparent surface representation of the main chain is shown. The ligand bound in the nucleotide-binding site is shown in stick representation. **a**, **c**, **e**, **g**, **j** Top and side views of unique PilT$^{Gm}$ crystal structures. O and C denote open and closed interfaces, respectively. **b**, **d**, **f**, **h**, **k** Correspond to the hexamers shown above and show a cross section of the closed- and open interfaces on the top and bottom panels, respectively. Select side chains discussed in the text are shown for reference. 2|$F_O$|−|$F_c$| maps are shown as a mesh with $I/\sigma(I)$ levels of 2 with the exception of (**d**) where 1.5 was used. **a**, **b** Crystal structure of methylated PilT$^{Gm}$ in the OCOCOC conformation. **c**, **d** Crystal structure of PilT$^{Gm}$ without added nucleotide in the OCOCOC conformation. **e**, **f** Crystal structure of PilT$^{Gm}$ washed with ATP prior to crystallization in the OCOCOC conformation. **g**, **h** Crystal structure of PilT$^{Gm}$ incubated with ATP during crystallization at pH 6.5 in the CCCCCC conformation. **i** In vitro ATP hydrolysis (mean ± SEM, $n = 2$) by PilT$^{Gm}$ from pH 5.5 to 9.0. **j**, **k** Crystal structure of PilT$^{Gm}$ incubated with ANP during crystallization in the CCOCCO conformation

crystals diffracted to 3.0 Å, and the structure was solved by molecular replacement with the entire hexamer present in the asymmetric unit (Fig. 1c). This structure was approximately $C_3$ symmetric with alternating open- and closed interfaces, similar to the methylated OCOCOC PilT structure. However, small deviations make this structure pseudo-$C_3$ symmetric. Although a nucleotide was not added, density consistent with ADP allowed the nucleotide with partial occupancy to be modeled in all nucleotide-binding sites (Fig. 1d). These nucleotides were likely carried over from *Escherichia coli* during protein purification. Nucleotide was absent in the methylated PilT$^{Gm}$ structure, possibly due to the lengthy methylation protocol or competition with sulfate during crystallization.

An isomorphous structure with high ADP occupancy was obtained by preincubating PilT$^{Gm}$ with Mg$^{2+}$ and ATP, then removing the unbound nucleotide prior to crystallization (Fig. 1e, f). This structure is consistent with the OCOCOC structure reflecting a post-hydrolysis ADP-bound conformation. The isomorphic low-occupancy and high-occupancy ADP OCOCOC PilT structures have an RMSD$^{C\alpha}$ of 0.6 Å per hexamer. The

RMSD$^{C\alpha}$ of these two structures with the methylated OCOCOC PilT structure is 1.9 Å per hexamer.

**PilT$^{Gm}$ also crystallizes in a $C_6$ symmetric conformation.** Modifying the protocol so exogenous ATP that was not removed prior to crystallization yielded distinct PilT$^{Gm}$ crystals that diffracted to 1.9 Å, the highest resolution to date for any hexameric PilT/VirB11-like family member. The structure was solved by molecular replacement with three protomers in the asymmetric unit. In the crystal, two nearly identical $C_6$ symmetric hexamers could be identified (Fig. 1g). Compared with the interfaces of PilB$^{Gm}$, all six interfaces in these hexamers are closed. This conformation is denoted CCCCCC.

Density consistent with Mg$^{2+}$ and surprisingly for an active ATPase ATP could be modeled in the active sites (Fig. 1h). The ribose moiety of ATP puckers in two alternate conformations consistent with the small number of direct protein contacts to the O2' of ATP (Fig. 1j). These conformations are consistent with C2' exo and C2' endo low-energy ATP ribose conformations[29]. In

addition, two ethylene glycol molecules from the cryoprotectant solution could be modeled in each packing-unit interface. The ethylene glycol was introduced after the crystals formed and bound to Arg-83 and Arg-278, next to the nucleotide-binding site. As these crystals formed only when the pH was less than or equal to 6.5, we hypothesised that PilT$^{Gm}$ may not have ATPase activity at acidic pH. When we assayed PilT$^{Gm}$ ATPase activity over a broad range of pH values, we could not measure ATPase activity below pH 6.5 (Fig. 1i). H230 of the HIS-box motif in PilT is predicted by Rosetta[30] to have a pKa of 6.5; thus, H230 deprotonation might be important for efficient ATP hydrolysis. The corresponding histidine in PilB coordinates the nucleotide γ-phosphate, but in the CCCCCC PilT structure, H230 is facing away from the γ-phosphate in each of the nucleotide-binding sites. We propose that the protonation state of H230 affects its preferred rotamer and thus the catalytic activity of PilT. Since H230 is conserved in all PilT/VirB11-like family members[18], this pH dependency for activity may be a conserved feature.

**PilT$^{Gm}$ also crystallizes in a PilB-like conformation**. To determine what conformation PilT$^{Gm}$ adopts above pH 6.5 with a non-hydrolyzable ATP analog, we also crystallized PilT$^{Gm}$ with Mg$^{2+}$ and ANP (adenylyl-imidodiphosphate or AMP-PNP) at pH 8. Unlike previous PilT$^{Gm}$ crystals that formed after 16 h and were stable for weeks, these crystals took a week to form and stayed crystalline for 2 days before dissolving. These crystals diffracted anisotropically to 4.1-, 6.7-, and 4.0-Å resolution along $a^\star$, $b^\star$, and $c^\star$ reciprocal lattice vectors, respectively. The structure was solved by molecular replacement, and a hexamer was present in the asymmetric unit (Fig. 1j). Despite the low resolution, density consistent with ANP could be modeled into four of the six nucleotide-binding sites (Fig. 1k). The density in the other two sites was consistent with ADP. It is possible, given the slow and transient crystallization, that ANP partially hydrolyzed, yielding a transient ADP/ANP mixture that facilitated formation of these particular crystals. In this case, decay to ADP is likely non-catalytic. This structure of PilT$^{Gm}$ is $C_2$ symmetric but distinct from that of the OOCOOC PilT$^{Aa}$ structure[22] (RMSD$^{C\alpha}$ 6.8 Å/hexamer). Surprisingly, the pattern of open- and closed interfaces between packing units was PilB-like: CCOCCO.

**ATP but not ADP binding correlates with closed interfaces**. As PilT$^{Gm}$ crystallized in multiple conformations and the resolution and quality of the electron density was sufficient to resolve the bound nucleotides, we looked for correlations between ADP or ATP/ATP-analog binding and the open- or closed interfaces. In the ADP-bound OCOCOC PilT$^{Gm}$ structures, the estimated occupancy of ADP was similar in the open- and closed interfaces (Fig. 1d, f). In the CCCCCC PilT$^{Gm}$ structure, all interfaces bound to ATP (Fig. 1j). In the CCOCCO PilT$^{Gm}$ structure, the four closed interfaces are bound to the ATP analog ANP, while the open interfaces appear to be bound to ADP (Fig. 1h). Thus, there is a correlation between bound ATP (or ATP analog) and closed interfaces, suggesting that ATP facilitates closure of interfaces in PilT$^{Gm}$. In contrast, there is no correlation between bound ADP and open and closed interfaces, suggesting that both open- and closed interfaces in PilT may have a similar affinity for ADP, and ADP may be insufficient to induce or maintain closure. This scenario contrasts with PilB$^{Gm}$ structures, where ADP is correlated with closed interfaces[18].

**Conserved interactions facilitate open or closed interfaces**. We established previously that in the closed interface of PilB$^{Gm}$, T411 contacts H420, and in the open interface, R455 contacts T411 (ref.[18]). To comprehensively define the interfaces in PilT$^{Gm}$, we

graphically plotted the open- and closed-interface contacts of the PilT$^{Gm}$ crystal structures, as well as previously published PilT structures (Fig. 2). For this analysis, a contact was defined as any atom (main chain or side chain) of a residue within 4 Å of any atom from another residue. To compensate for imperfect rotamers in low-resolution structures, the definition of contacts for this analysis was expanded to also include main-chain atoms in one residue within 8 Å of a main-chain atom from another residue.

Despite disparate hexameric conformations, the PilT structures make fairly consistent open- and closed-interface contacts. Residues in the closed interface that contact one another include P271–D32, G277–D32, H230–T221, T221–T133, R195–D161, and D164–R81. P271–D32 and H230–T221 are specific to the closed interface. Residues in the open interface that consistently contact one another include G339–Y259, G180–Q59, L269–T221, R195–T133, T221–T133, E178–S73, I163–S73, and D164–R81. G339–Y259, G180–Q59, and L269–T221 are specific to the open interface. Many of these residues are part of the conserved HIS (T221 through H230) and ASP (E160 through E164) box motifs[31,32]. Involvement in the open and closed interfaces explains why several residues in these motifs are conserved, even though only H230 and E164 contact ATP or magnesium, respectively[18,22]. The contacts observed in the open and closed interfaces are similar to those found in the PilB structure: T221 contacts H230 in the closed interface (T411 and H420 in PilB$^{Gm}$), while T221 contacts L269 (T411 and R455 in PilB$^{Gm}$) in the open interface.

Based on this information, closed- and open interfaces can be easily distinguished by measuring the Cα distance between inter-chain residues T221–H230 and T221–L269. In the closed interfaces of PilT$^{Gm}$ structures, the mean Cα distances for T221–H230 and T221–L269 are $5.7 \pm 0.3$ Å (mean ± SD) and $13.7 \pm 0.8$ Å, respectively. In the open interfaces of PilT$^{Gm}$ structures, the mean Cα distances for T221–H230 and T221–L269 are $12.7 \pm 1$ Å and $8.6 \pm 0.4$ Å, respectively. The T221–H230 and T221–L269 distances are closer in the closed- and open interfaces, respectively, permitting easy interface classification.

In subsequent analyses, we used a cutoff of <9 Å for the T221–H230 distance and >12 Å for T221–L269 to define the closed interface. To define the open interface, we used a cutoff of >12 Å for T221–H230 and <11 Å for T221–L269. The T221–H230 and T221–L269 distances were measured across representative PilT/VirB11-like structures, to identify their closed- and open interfaces (Supplementary Table 1). The mean T221–H230 and T221–L269 backbone distances in PilT/VirB11-like structures are, respectively, $6 \pm 1$ Å and $16 \pm 3$ Å in closed interfaces, and $16 \pm 4$ Å and $9 \pm 1$ Å in open interfaces.

**PilT/VirB11-like proteins adopt six conformational states**. Characterization of the open and closed interfaces in all PilT/VirB11-like family member structures allowed most to be placed into one of five different states based on their conformation: CCOCCO, OCOCOC, CCCCCC, OOOOOO, or OOCOOC (Fig. 3). These states reflect all possible arrangements of open- and closed interfaces that maintain rotational symmetry. A GspE structure (PDB 4KSS) and all VirB11 crystal structures (PDB 2PT7, 2GZA, 1NLY, 1NLZ, and 1OPX) are CCCCCC (Fig. 3a). In addition to the ATP-bound PilT$^{Gm}$ structure described herein, the PilT structures from *Pseudomonas aeruginosa* (PilT$^{Pa}$) are also in the CCCCCC conformation (PDB 3JVV and 3JVU) (Fig. 3a). The three OCOCOC PilT structures described here are the only examples of PilT in this conformation determined to date (Fig. 3b). A FlaI and a DotB structure (PDB 4II7 and 6GEB,

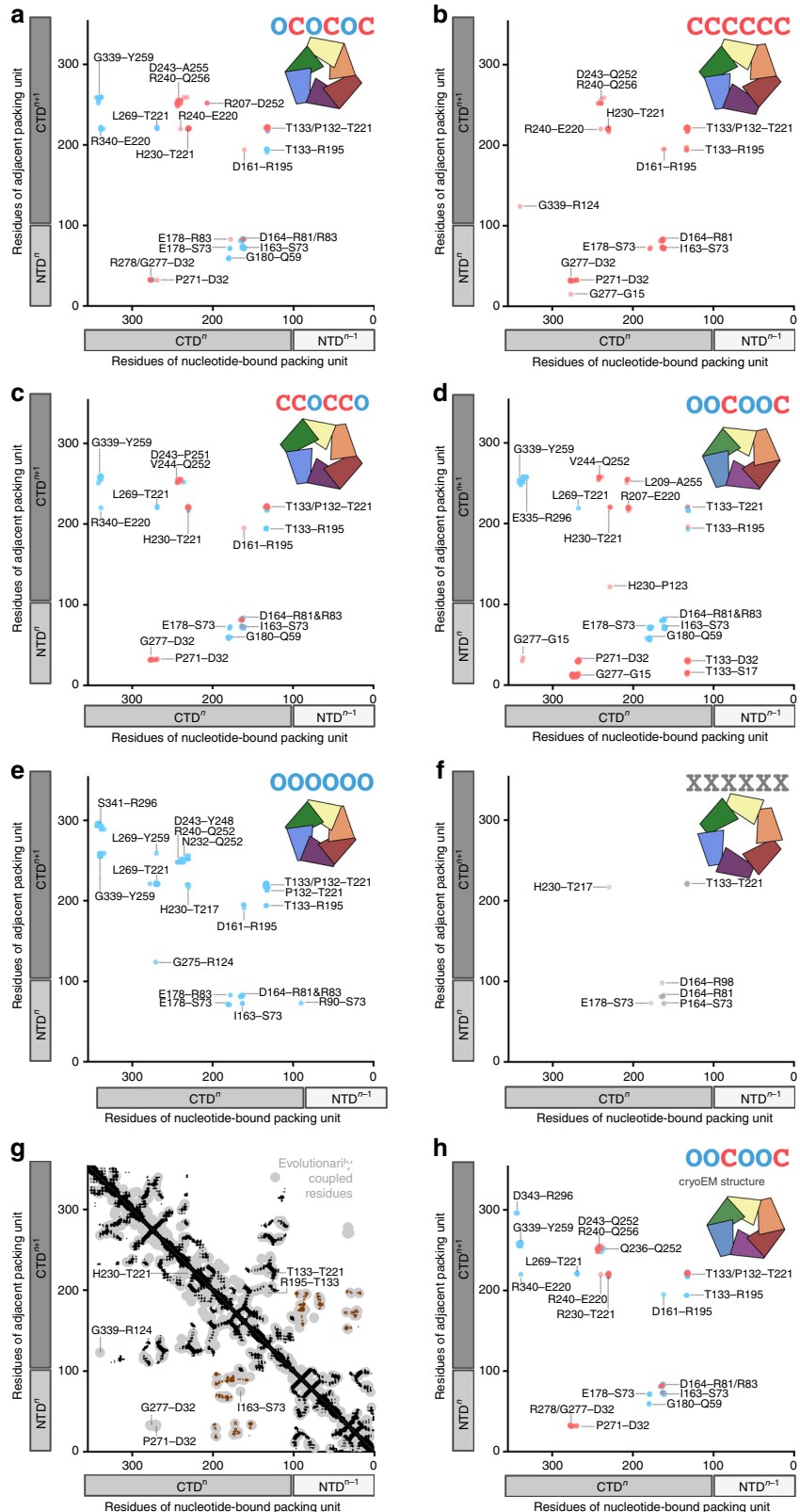

respectively), as well as two Archaeal GspE2 structures (2OAP and 2OAQ) have OCOCOC conformations (Fig. 3b). Other GspE, FlaI, and DotB structures (PDB 4KSR, 4IHQ, and 6GEF) fall into the CCOCCO class (Fig. 3c). All available PilB structures are CCOCCO (Fig. 3c). Our CCOCCO PilT$^{Gm}$ structure is the only example of PilT in this conformation to date (Fig. 3c). PilT$^{Aa}$ is the only example of the OOCOOC conformational state (PDB

2GSZ) (Fig. 3d). Similarly, only PilT4 from *G. sulfurreducens* (PilT$^{Gs}$) exhibits an OOOOOO conformation (Fig. 3e). This classification scheme suggests that PilT and PilT/VirB11-like family member crystal structures have a high fidelity for open or closed interfaces and rotational symmetry.

Cross-referencing PilT structural classification with packing-unit interface contacts (Fig. 2) revealed that there are contacts

**Fig. 2** Contact maps for packing-unit interfaces reveal similarities between distinct PilT conformations. Inter- and intra-chain contacts between packing units color coded based on closed (red) or open (blue) interface; the x-interface is shown in gray. For reference, the linear domain architecture of individual packing units is shown as box cartoons beside the axes with each chain shaded distinctly. To minimize confusion when comparing between species the residue labels represent the corresponding residue in PilT[Gm]. **a** OCOCOC conformation PilT[Gm] crystal structure (PDB 6OJY). **b** CCCCCC conformation PilT[Gm] crystal structure (PDB 6OJX). **c** CCOCCO conformation PilT[Gm] crystal structure (PDB 6OKV). **d** OOCOOC conformation PilT[Aa] crystal structure (PDB 2GSZ). **e** OOOOOO conformation PilT[Gs] crystal structure (PDB 5ZFQ). **f** XXXXXX conformation PilT[Aa] crystal structure (PDB 2EWV). **g** Evolutionarily coupled residues in PilT (light gray) are compared with the structure of PilT[Gm]. Tertiary structure contacts (not including intra-chain interactions between packing units) are noted in black. N2D$^n$ to CTD$^{n+1}$ contacts (that create individual packing units) are noted in brown. Open- or closed-interface contacts identified in (**a–f**) are labeled if they overlap with evolutionarily conserved residue pairs that are not clearly accounted for by the tertiary structure or N2D$^n$ to CTD$^{n+1}$ contacts. **h** OOCOOC conformation PilT[Gm] cryoEM structure (PDB 6OLL)

unique to some conformational states. For example, the G339–R124 and G275–R124 contacts are specific to the CCCCCC and OOOOOO PilT conformational states, respectively (Fig. 2b, e). There are many contacts unique to the OOCOOC PilT structure (Fig. 2d), though this may be a result of the evolutionary distance of PilT[Aa] from those of Proteobacteria. There are no contacts unique to the CCOCCO and OCOCOC PilT[Gm] structures (Fig. 2a, c).

PilT[Aa] was crystallized in a distinct $C_6$ symmetric conformation bound to ATP (PDB 2EWW) or ADP (PDB 2EWV) (Fig. 3f). The two structures can be considered isomorphic (RMSD$^{C\alpha}$ 0.6 Å/hexamer) and are distinct from other PilT/VirB11-like family member structures. In these structures, there is almost no interface between packing units and they appear to be held in place by crystal contacts, suggesting that this conformation may be uncommon outside of a crystal lattice. The distances of T221–H230 and T221–L269 are atypically large (11 and 14 Å, respectively) (Supplementary Table 1). These distances suggest that the packing units adopt neither a closed- nor open-interface and thus we refer to it as an X-interface. Thus, these $C_6$ symmetric PilT[Aa] structures represent a distinct conformational state: XXXXXX. The H242–R229 salt bridge is a major constituent of the X-interface; these residues are H230 and T217, respectively, in PilT[Gm] (Fig. 2f). As R229 in PilT[Aa] is not conserved in PilT[Gm], the X-interface or XXXXXX conformation may not be critical to function. Supporting this hypothesis, it was suggested that these XXXXXX PilT[Aa] structures are not in a conformation that could facilitate ATP hydrolysis[22].

**Residues in the closed interface are evolutionarily coupled.** One explanation for the perceived heterogeneity in structures of PilT and PilT/VirB11-like family members is that the proteins are conformationally heterogeneous and the crystallization process selects for just one of many possible conformations. To probe this possibility, we used the EVcouplings server[33,34] to identify residues that coevolve in PilT (Fig. 2g). Residues that contact one another — and are needed for biological function — tend to be evolutionarily coupled, and therefore this analysis can be used to independently validate structural analyses[33,34].

The evolutionarily coupled residues of PilT were consistent with its tertiary structure contacts, as well as the contacts that support the packing unit. There were also evolutionarily coupled residues clustered around G277–D32, P271–D32, H230–T221, T221–T133, R195–T133, and G339–R124 consistent with the closed interface, and I163–S73 consistent with the open or closed interface. This analysis unambiguously demonstrates phylogenetic conservation of the closed interface across PilT orthologs. The G339–R124 contact is specific to the CCCCCC PilT[Gm] structure, specifically validating the biological relevance of this conformation.

The quality of sequence alignments is worse near the C terminus of PilT, and thus the power to find evolutionarily conserved residues is lower at the C terminus. Also, some open-

and closed-interface contacts overlap graphically with tertiary or packing-unit-forming contacts, obscuring their identification in this type of analysis. Thus, while this analysis validates the CCCCCC PilT structure and closed interface, other non-crystallographic techniques are required to test the biological relevance of particular conformations.

**2D cryoEM of PilT[Gm] reveals conformational heterogeneity.** To examine what conformation(s) PilT can adopt in a noncrystalline environment we used cryoEM. We found that PilT[Gm] adopted preferred orientations on the EM grid with its symmetry axis normal to the air–water interface (top views), preventing calculation of 3D maps from untilted images. Nonetheless, comparison of the shape, size, and internal features of the 2D class average images with projections of PilT crystal structures (Fig. 4a) allowed for unambiguous assignment of 2D class averages into one of the six defined PilT conformational states. In the absence of the nucleotide, the 2D class averages of PilT[Gm] were a mixture of the OCOCOC and OOCOOC conformational classes in a 45:55 ratio (Fig. 4a).

In the presence of 0.1 mM ADP and 1 mM ADP, the ratio of the particles in the OCOCOC and OOCOOC states shifted from 45:55 to 54:46 and 71:29, respectively (Supplementary Fig. 2). Thus, adding ADP favors the OCOCOC conformation, consistent with crystallographic OCOCOC structures of PilT[Gm] with ADP bound in every interface. There may be an unoccupied nucleotide-binding site available in the OOCOOC conformation that upon binding ADP converts to the OCOCOC conformation. This hypothetical unoccupied nucleotide-binding site might prefer to bind a different nucleotide such as ATP, since mM quantities of ADP were required to favor the OCOCOC conformation.

**PilT[Pa] also has conformational heterogeneity.** PilT[Pa] has been crystallized in the CCCCCC conformation (Fig. 3a) both in the absence of nucleotide and in the presence of an ATP analog[24]. We hypothesized that purified PilT[Pa] may also exhibit the OCOCOC or OOCOOC conformation in the absence of nucleotide in a noncrystalline environment. Thus, cryoEM analysis was performed with purified PilT[Pa]. Analysis of the 2D class averages were — like PilT[Gm]— consistent with the OCOCOC and OOCOOC conformations (Fig. 4a). The particle distribution ratio between OCOCOC and OOCOOC class averages was 63:37.

**3D cryoEM validates coexisting OOCOOC and OCOCOC structures.** To confirm our interpretation of the 2D class averages, we tilted the specimen[35] to obtain sufficient views of the hexamers to calculate 3D maps. CryoEM specimens of PilT[Gm] without nucleotide were tilted by 40° during data collection. Using heterogeneous refinement in cryoSPARC[36] and per-particle determination of contrast transfer function parameters[37], two distinct maps could be obtained from the data

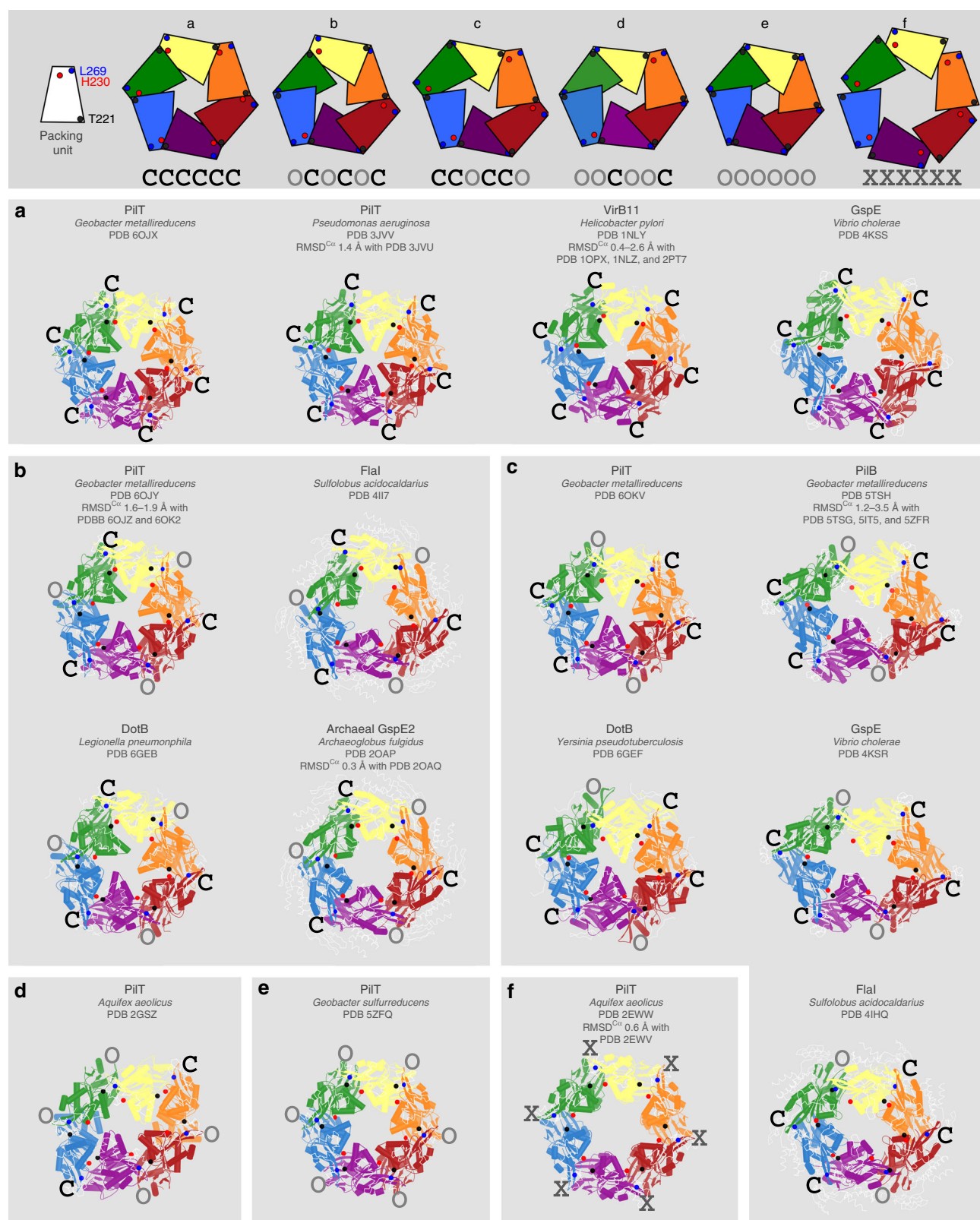

without enforcing symmetry: one ~$C_3$ map and one ~$C_2$ map, both at ~4.4-Å resolution. The particle distribution ratio was 48:52 between $C_3$ and $C_2$ maps, respectively, similar to that obtained by 2D classification, validating the 2D conformation assignments. Applying their respective symmetries during refinement yielded 4.0 and 4.1 Å resolution maps, respectively (Fig. 4c, d).

Molecular models could be built into these maps by fitting and refining rigid packing units of the PilT$^{Gm}$ crystal structures (Fig. 4c, d). Assessment of local resolution suggests that the non-surface-exposed portions of the map are at higher resolution than the rest of the complex (Supplementary Fig. 3). No density consistent with nucleotide was identified in these structures; presumably a nucleotide that was potentially carried over from

**Fig. 3** All PilT/VirB11-like family member structures can be divided into one of six unique conformations. Structures are shown as cartoons with individual packing units (N2D$^n$ plus CTD$^{n+1}$) uniquely colored. Black spheres, the α-carbons of the residues that align with T221 from PilT$^{Gm}$. Red spheres, the α-carbons of the residues that align with H230 from PilT$^{Gm}$. Blue spheres, the α-carbons of the residues that align with L269 from PilT$^{Gm}$. Top, block cartoons of PilT/VirB11-like family hexamer conformations. **a–f** To highlight similarities between conformations, structural elements — like domains, α-helices, or β-sheets — that are not well conserved across all PilT/VirB11-like family members are shown as thin white ribbons. The hexamers are labeled with their protein name, followed by the species of origin, followed by their PDB identifier, and finally the PDB identifiers of similar structures with the range of RMSD$^{Cα}$ (over the full hexamer) of these structures aligned with the shown structure. The interface between packing units is annotated as determined by Using the T221, H230, and L269 distances in Supplementary Table 1. **a** PilT/VirB11-like proteins with the CCCCCC conformation. **b** PilT/VirB11-like proteins with the OCOCOC conformation. **c** PilT/VirB11-like proteins with the OOCOOC conformation. **d** The only PilT/VirB11-like protein with the OOCOOC conformation. **e** The only PilT/VirB11-like protein with the OOOOOO conformation. **f** The only PilT/VirB11-like protein with the XXXXXX conformation

the *E. coli* expression system is present at too low an occupancy to be observed. The model built into the $C_3$ symmetric map is consistent with the OCOCOC PilT structure (RMSD$^{Cα}$ 1.3 Å/ hexamer). Before symmetry was applied to this map, it more closely matched the methylated $C_3$ symmetric structure of PilT than the pseudo-$C_3$ symmetric structures, suggesting that the slight asymmetry of the latter is a crystallographic artifact. The model built into the $C_2$ symmetric map was not consistent with any PilT$^{Gm}$ crystal structure. Annotation of its packing-unit interfaces revealed that it has an OOCOOC conformation, consistent with the PilT$^{Aa}$ crystal structure (Supplementary Table 1). Thus, the cryoEM structures confirm that the OOCOOC and OCOCOC conformations observed for PilT$^{Aa}$ and PilT$^{Gm}$, respectively, were not crystal artifacts. Further, these maps suggest that available crystal structures have oversimplified our view of PilT/VirB11-like family members as they do not capture the multiple stable conformations accessible in a given condition.

While the OOCOOC PilT$^{Gm}$ cryoEM structure validates the conformation of the OOCOOC PilT$^{Aa}$ crystal structure, the two are distinct (RMSD$^{Cα}$ of 6.4 Å/hexamer), consistent with the evolutionary distance between species. Analyzing the packing-unit interfaces of the OOCOOC PilT$^{Gm}$ cryoEM structure reveals that they are nearly identical to the interfaces in the PilT$^{Gm}$ CCOCCO and OCOCOC crystal structures (Fig. 2h).

**CryoEM of PilT$^{Gm}$ with ATP reveals CCCCCC conformation.** Since PilT hydrolyzes ATP slowly and cryoEM samples can be frozen within minutes of sample preparation, we opted to determine the conformation of PilT$^{Gm}$ incubated briefly with ATP. In these conditions, the top-view 2D class averages of PilT$^{Gm}$ corresponded only to the CCCCCC conformational class, consistent with the ATP-bound CCCCCC PilT crystal structure (Fig. 4a). A small minority of 2D class averages appeared to be tilted- or stacked side views, permitting 3D map construction. Only one map with ~$C_6$ symmetry could be constructed and applying $C_6$ symmetry during refinement resulted in a 4.4 Å resolution map (Fig. 4e). The molecular model built from this map is consistent with the CCCCCC crystal structure (RMSD$^{Cα}$ 0.6 Å) and density in the nucleotide-binding sites is consistent with ATP (Fig. 4e).

In an attempt to reproduce the conditions that we postulated led to the CCOCCO PilT crystal structure, PilT$^{Gm}$ was incubated with mixtures of ATP and ADP. In the presence of 1 mM ATP and ADP, or 1 mM ATP and 0.1 mM ADP, only class averages consistent with the CCCCCC conformation could be identified (Supplementary Fig. 2). This analysis does not support the reproducibility of the CCOCCO PilT conformation in a noncrystalline environment, nor the OOOOOO or XXXXXX PilT conformations, which were not identified in any condition. The cryoEM experiments suggest that in the absence of its protein-binding partners in vitro, at approximately physiological

ATP and ADP concentrations, PilT$^{Gm}$ is predominantly found in the CCCCCC conformation.

**CryoEM analysis of PilB$^{Gm}$ consistent with CCOCCO conformation.** During the course of our studies, an 8-Å cryoEM structure of PilB from *T. thermophilis* (PilB$^{Tt}$) was published that revealed a CCOCCO conformation in a noncrystalline environment[27]. No conformational heterogeneity was reported[27]. To determine whether this homogeneity was observed in a Proteobacteria PilB, we performed cryoEM analysis of PilB from *G. metallireducens* (PilB$^{Gm}$) in the absence of nucleotide. Projection of the PilB CCOCCO crystal structure revealed that the PilB$^{Gm}$ top-view 2D class averages were consistent with the CCOCCO conformation (Fig. 4b). A 3D map was calculated at ~7.8-Å resolution (Fig. 4f), and the model built into this map is also consistent with the CCOCCO PilB structure (RMSD$^{Cα}$ 2.3 Å/ hexamer, PDB 5TSG). Thus, cryoEM analysis reveals that PilB preferentially adopts the CCOCCO conformation in multiple species. This is in contrast to PilT$^{Gm}$ and PilT$^{Pa}$ that both adopt OOCOOC and OCOCOC conformations in similar conditions. These results show that the preferred conformation(s) are conserved within — but not between — PilT/VirB11-like subfamilies, consistent with distinct conformation preferences facilitating PilB-like or PilT-like functions. It should be noted that the N-terminal domain of PilB$^{Gm}$, known as the N1D, MshEN, or GSPII domain, was not observed in the 3D map although its presence was confirmed by trypsin digest followed by mass spectrometry (98% coverage from the His tag to the C terminus). It may be that in the absence of its binding partners, this domain is disordered relative to the core motor domains of PilB$^{Gm}$.

**CCCCCC and OOCOOC or OCOCOC conformations are essential.** The cryoEM structures of PilT$^{Gm}$ and PilB$^{Gm}$ were determined in the absence of other components of the T4aP system. To explore the functional importance of these PilT conformations in vivo, we introduced mutations targeting the packing-unit interface into *P. aeruginosa*, a model organism for studying PilT function. From our analysis of key contacts (Fig. 2), we mutated residues predicted to alter packing-unit interfaces and thus overall conformations. As controls, we mutated the catalytic glutamate (E204A) and the γ-phosphate coordinating HIS-box histidine (H229A), which eliminates twitching motility in *P. aeruginosa*[11]. E204A and H229A mutants lost twitching motility and accumulated extracellular PilA, while E204A also led to PO4 phage resistance, consistent with a retraction defect (Fig. 5a).

In PilT$^{Gm}$ structures, residue R240 participates in the closed interface, E220, D32, and D243 are at both open- and closed interfaces, and F259, R296, and Q59 are at the open interface (Fig. 2). In PilT$^{Pa}$ these correspond to R239, E219, D31, E258, D242, R294, and K58, respectively. Mutation of most of these residues abrogated twitching motility, increased extracellular

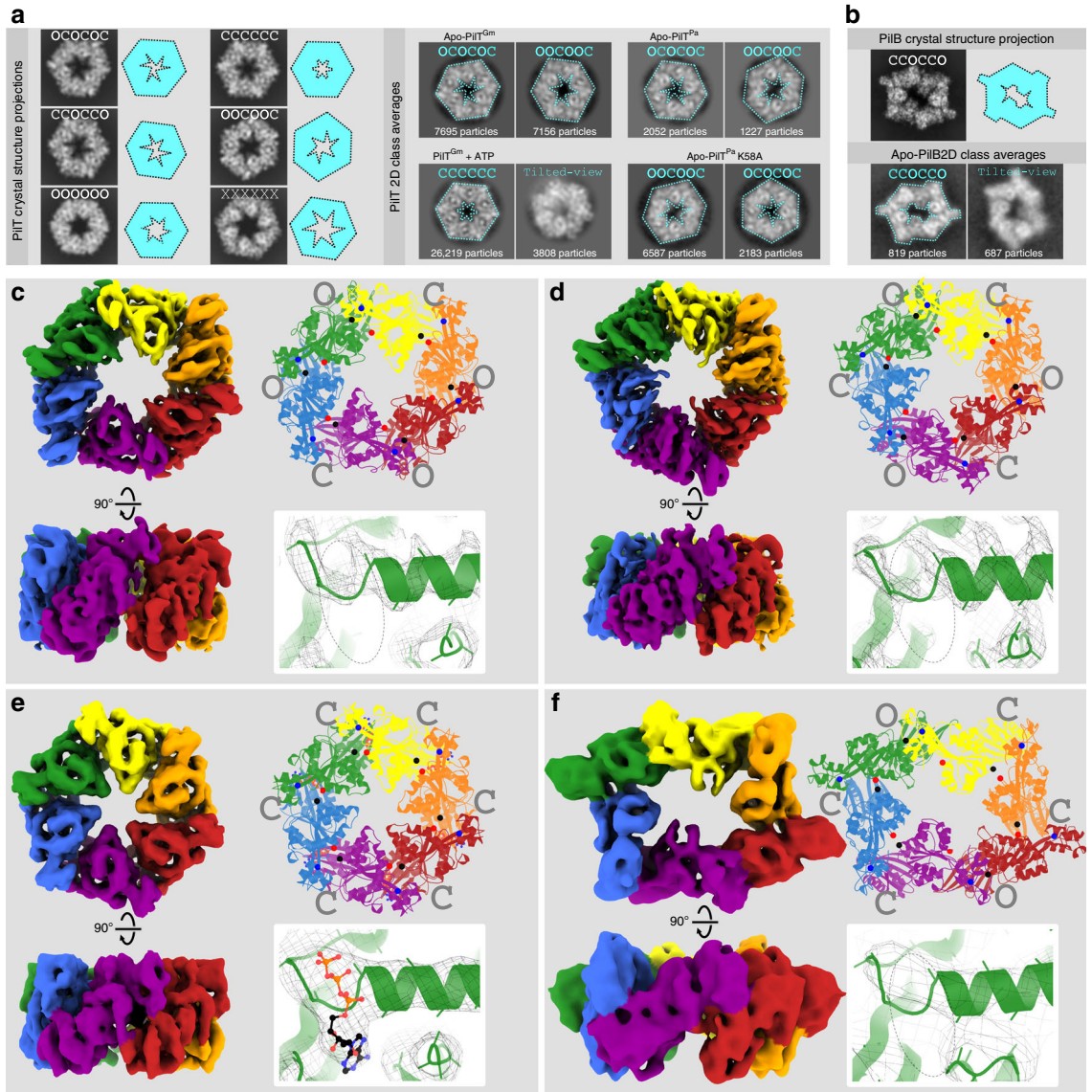

**Fig. 4** CryoEM structure analysis reveals the conformation preferences of PilT and PilB. **a** PilT crystal structures were used to simulate top-view cryoEM projections (left). For this analysis, hexamers representative of OCOCOC, CCCCCC, CCOCCO, OOCOOC, OOOOOO, and XXXXXX conformations were extracted from the corresponding PDB coordinates: 6OJY (this study), 6OJX (this study), 6OKV (this study), 6OLL (this study), 5ZFQ, and 2EWV, respectively. Simplified cartoons of these conformations are shown beside the corresponding projection (cyan cartoons). Images of representative 2D class averages (right) are shown for apo-PilT^Gm particles, particles of PilT^Gm preincubated with 1 mM ATP, apo-PilT^Pa particles, and apo-PilT^Pa particles with the K58A mutation. Using the simulated projections, the 2D class averages (right) were annotated with their predicted conformation (cyan font), and the simplified cartoon outline is overlaid on the 2D class average (cyan dotted line). For scale, the gray squares showing the 2D class averages are 16-nm wide. **b** The hexamer from a PilB^Gm crystal structure (5TSG) was extracted and used to simulate a cryoEM projection (top). Images of representative 2D class averages from apo-PilB^Gm particles are also shown (bottom). **c–f** Locally sharpened cryoEM maps colored as in Fig. 1 by packing units (left); the top view of models built into these maps (cartoons, top right) with spheres shown for the α-carbons of residues that align with T221, H230, and L269 as in Fig. 3 (black, red, and blue spheres, respectively), as well as the open- and closed interfaces (gray), and an expanded view of the Walker A motif (bottom right) showing the quality of the sharpened maps (mesh) and the presence or absence of nucleotides. A dotted oval outline shows the location of the nucleotide-binding site. **c** CryoEM map of PilT in the OOCOOC conformation to 4.1-Å resolution determined in the absence of added nucleotide. **d** CryoEM map of PilT in the OCOCOC conformation to 4.00-Å resolution also determined in the absence of added nucleotide. **e** CryoEM map of PilT preincubated with 1 mM ATP in the CCCCCC conformation to 4.4-Å resolution. **f** CryoEM map of PilB in the CCOCCO conformation to 7.8-Å resolution determined in the absence of added nucleotide

PilA, and led to PO4 phage resistance (Fig. 5a). These data suggest that the XXXXXX PilT conformation, lacking open- or closed interfaces, is insufficient for PilT function. Likewise, these experiments imply that neither the CCCCCC nor OOOOOO PilT conformations — lacking open or closed interfaces, respectively — are sufficient for PilT function. The E219K and D31K mutants had reduced stability or expression, complicating their interpretation (Fig. 5a). The side chains of these residues participate in the closed interface but not the open interface. Curiously, despite the instability of D31K and the corresponding accumulation of PilA, D31K had significantly increased twitching motility and was partially susceptible to PO4 phage infection (Fig. 5a). One interpretation of this phenotype is that PilT misfolded in most bacteria expressing the D31K mutant, leading

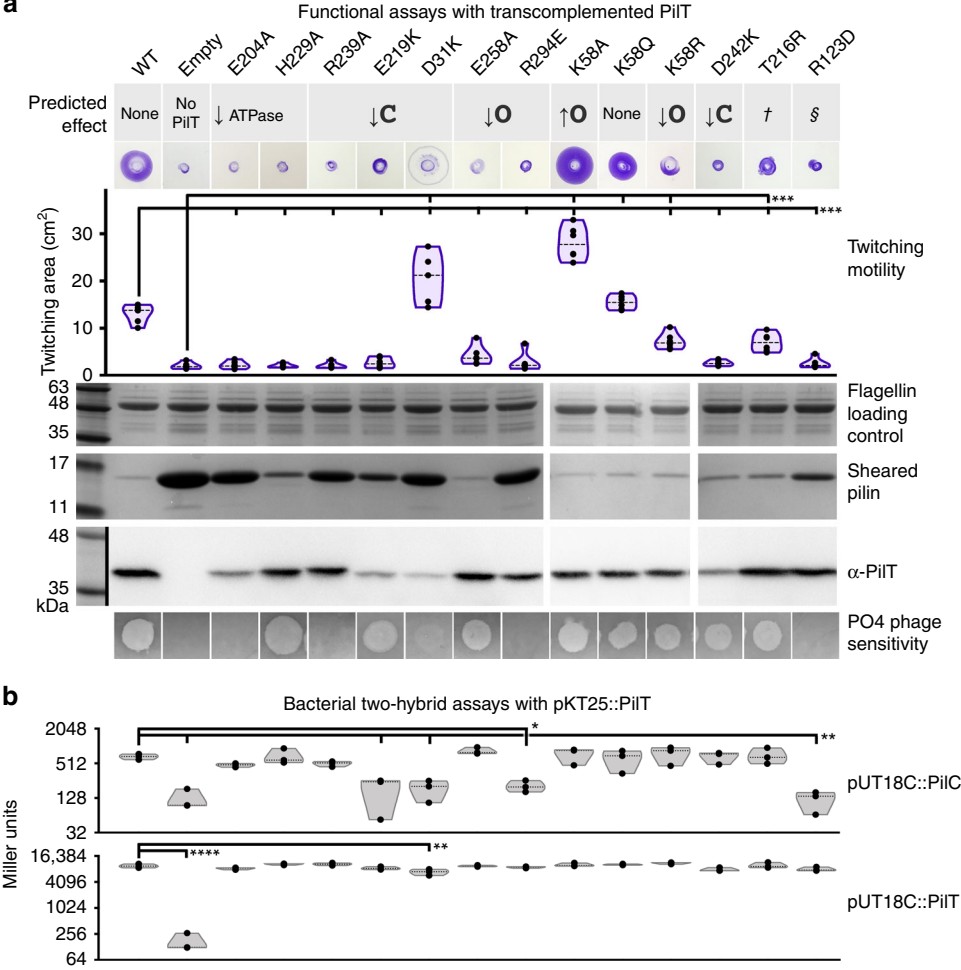

**Fig. 5** Multiple conformations and both open- and closed interfaces are essential for PilT in vivo function. **a** A *P. aeruginosa* PAO1 PilT deletion mutant was transcomplemented with wild type (WT), a vector control (Empty), and various site mutants. Their predicted effects are indicated. Dagger symbol, the T216R mutant is predicted to ablate the OOOOOO conformation. Section sign symbol, the R123D mutant is predicted to reduce the stability of the OOOOOO or CCCCCC conformations. Twitching motility was assayed by crystal violet staining (purple zones), and the areas covered by the bacteria were measured (purple violin plot, $n = 6$). Mean twitching areas were compared with the one-way ANOVA test, noted with black lines; reported p-values were less than 0.001 (***). Transcomplemented bacteria were grown on a surface; pilins and flagellins were sheared from the surface of the bacteria and assayed by SDS-PAGE analysis, while whole-cell lysates were probed with an anti-PilT antibody (middle). PO4 phage was spotted onto double-layer agar inoculated with the transcomplemented bacteria; transilluminated images of the resulting zones of lysis are shown (bottom). **b** Quantitative bacterial two-hybrid analysis of PilT[Pa] mutants. The mutations in **a** were introduced into the pKT25::PilT[Pa] two-hybrid construct. The adenylate cyclase mutant, *E. coli* BTH101 was then co-transformed with that construct and either pUT18C::PilC[Pa] (top gray violin plot, n = 3) or pKT25::PilT[Pa] (bottom gray violin plot, n = 3) and interactions assessed by measuring β-galactosidase activity in Miller units. The mean Miller units were compared with the one-way ANOVA test, noted with black lines; reported p-values were less than 0.02 (*), 0.01 (**), or 0.0001 (****). Source data are provided as a Source Data file

to accumulation of extracellular PilA, but in a subpopulation of bacteria the D31K mutant protein folded properly and unexpectedly facilitated increased twitching motility. Alternatively, it is possible that the D31K mutation reduced binding of the antibody used to detect PilT. The R294E mutation, predicted to decrease the stability of the open interface, decreased twitching motility, pilin accumulation, and phage resistance, consistent with a pilus depolymerization defect. In contrast, the E258A mutation, also predicted to decrease the stability of the open interface, decreased twitching motility but allowed approximately wild-type levels of pilin accumulation and phage susceptibility, consistent with a pilus polymerization defect, or more plausibly a moderate defect in PilT function that did not cause an obvious overabundance of extracellular pilin.

The K58Q mutant of PilT[Pa] had wild-type twitching motility, extracellular pilin accumulation, and was sensitive to PO4 phage, consistent with this residue being glutamine in wild-type PilT[Gm].

Surprisingly, the K58A mutant had twofold increased twitching motility and decreased levels of extracellular PilA indicative of hyperretraction, while the conservative K58R mutation reduced twitching motility (Fig. 5a).

Mutations were also introduced at residues that are important for particular conformations. The R124 residue in PilT[Gm] stabilizes the CCCCCC and OOOOOO conformations, as it forms a salt bridge with the backbone carbonyl of G339 and G275, respectively (Fig. 2). R124 in PilT[Gm] aligns with R123 in PilT[Pa]. The R123D mutation in PilT[Pa] eliminated twitching motility, prevented phage infection, and led to accumulation of extracellular pilins, consistent with a retraction defect (Fig. 5a). This result suggests that either the CCCCCC or OOOOOO PilT conformation is essential for retraction. In the OOOOOO conformation, T217 forms a polar interaction with H230 (Fig. 2) and its mutation to arginine is predicted to eliminate that conformation. T217 in PilT[Gm] aligns with T216 in PilT[Pa]. The

T216R mutant of PilT[Pa] had slightly decreased twitching motility, and wild-type pilin accumulation and phage infection (Fig. 5a), suggesting that OOOOOO conformation plays a minor role in PilT retraction. These results imply that the CCCCCC conformation is necessary for PilT function. In combination with the mutations targeting open and closed interfaces (above), these results indicate that the CCCCCC conformation and at least one open and closed interface containing conformation are essential for PilT function. This open and closed interface containing conformation is likely OOCOOC or OCOCOC, as they were the only PilT conformations with open and closed interfaces reproduced by cryoEM analysis.

**Hypertwitching K58A mutation favors OOCOOC conformation.** Based on the corresponding residues in the open interface of PilT[Gm], K58 is predicted to be near residues H179 and R180 in the open interface of PilT[Pa]. This proximity could cause electrostatic repulsion at the open interface. Thus, a K58A mutation may stabilize the open interface and favor conformations with more open interfaces. To test this hypothesis, cryoEM analysis was performed on purified K58A PilT[Pa]. The ratio of the particles in the OCOCOC and OOCOOC classes shifted from 63:37 for wild-type PilT[Pa] (Fig. 4d) to 25:75 for the K58A mutant (Fig. 4e). This result is consistent with the K58A mutation stabilizing the open interface and thus favoring the OOCOOC conformation versus the OCOCOC conformation. That a single point mutation can shift the ratio between the OOCOOC and OCOCOC conformations is consistent with different ratios of those conformations in PilT[Gm] and PilT[Pa]; K58 is a glutamine in PilT[Gm], for example.

**CCCCCC and the open interface are necessary for binding PilC.** We hypothesized that the observed retraction defects reflected the inability of some PilT mutants to adopt a conformation compatible with PilC binding. To test this hypothesis, we used bacterial two-hybrid analysis (BACTH) to quantify the interaction between PilT mutants and PilC (Fig. 5b). BACTH has been used previously to demonstrate an interaction between PilT and PilC[38]. Each PilT mutant was capable of homomeric interactions consistent with correct protein folding, with the exception of the D31K mutant, consistent with its putative stability defect (Fig. 5b). The open-interface-targeting R294E mutant and CCCCCC-targeting R123D mutant had reduced PilC interactions (Fig. 5b). These residues are not in the pore of PilT and thus unlikely to be important for directly contacting PilC, although confirmation of this hypothesis awaits a PilT–PilC co-structure. Accordingly, these results are consistent with CCCCCC and an open-interface-containing conformation being important for binding PilC.

**Discussion**
Here, we demonstrate the conformational heterogeneity of PilT. This protein can adopt conformations consistent with all PilT/VirB11-like family member conformational states defined here. We present several unique PilT[Gm] crystal structures and demonstrate that multiple conformations of PilT[Gm] and PilT[Pa] coexist in solution. We show that specific PilT conformations are important for in vivo function and interaction with PilC. By extrapolation, these findings have major ramifications for interpretation of other PilT/VirB11-like crystal structures, which have individually been used to suggest idiosyncratic molecular mechanisms. Based on our ability to clearly categorize all PilT/VirB11-like family members, we predict that PilT/VirB11-like family members of T4P-like systems operate with a common mechanism. The T4SS lacks a PilC-like inner-

membrane platform protein, so VirB11 or DotB may have distinct mechanisms.

This study unifies the structural description and analyses across PilT/VirB11-like family members. We found that the inter-chain distances between T221–H230 and T221–L269 can be used to easily and quantitatively define open and closed interfaces. This simple definition enables the conformational state of the hexamer to be defined and would easily be missed if only individual chains are annotated. Given the conservation of these inter-chain distances, we predict that these residues have functional significance. The catalytic glutamate E204 is thought to abstract a proton from a water molecule for subsequent hydroxyl nucleophilic attack of the γ-phosphate of ATP[18]. Given its location adjacent to E204, H230 may then abstract this proton and shuttle it to T221 in the closed interface. From T221, the proton could be passed directly or indirectly via T133 to the recently hydrolyzed inorganic phosphate. The requirement for T221 from an adjacent packing unit for proton shuttling would prevent efficient ATP hydrolysis in the open interface prior to its closure. Consistent with this proposed mechanism, we found that PilT[Gm] ATPase activity is pH sensitive in the range consistent with histidine protonation, and that in PilT[Gm] structures H230 faces away from the nucleotide-binding site in most open interfaces but toward the nucleotide-binding site in most closed interfaces.

Our highest resolution crystal structure of a hexameric PilT/VirB11-like family member determined to date also revealed that the ribose moiety of ATP can adopt multiple conformations due to the lack of interactions with its O2' hydroxyl. It is thus not surprising that ATP analogs with fluorophores attached at the O2' position bind PilT/VirB11-like family members[39,40]. We also found two ethylene glycol molecules in the packing-unit interface, suggesting that rationally designed small molecules could target this interface. Targeting the nucleotide-binding site or packing-unit interface to inhibit ATPase activity in T4P-like systems may have therapeutic value as these systems play a major role in virulence for many pathogens. Indeed, there are two recent reports of small-molecule inhibitors of the *Neisseria meningitidis* T4aP that target pilus polymerization and depolymerization dynamics to reduce virulence[41,42]. One of these drugs targets PilB directly[41], although whether this drug binds the nucleotide-binding site or packing-unit interface is not yet clear.

Our cryoEM maps establish the coexistence of both OOCOOC and OCOCOC conformational classes in PilT in the absence of a nucleotide or with added ADP. We also identified the CCCCCC conformation in the presence of ATP or approximately physiological concentrations of ATP and ADP. Based on these analyses, we propose a model explaining how the conformation of PilT — and probably other PilT/VirB11-like family members — changes in vitro depending on the nucleotides present (Fig. 6b). We also showed by cryoEM analysis, in accordance with the recently determined PilB[Tt] structure[27], that PilB[Gm] uniquely adopts the CCOCCO conformation in a noncrystalline state. Although the structures of these proteins were determined in isolation, the conformations observed are likely biologically relevant, as PilB and PilT are only intermittently engaged with PilC and the T4aP machinery. No core T4aP proteins are unstable in the absence of PilB or PilT[43], cryoET analysis of the T4aP in *M. xanthus* shows a significant portion of T4aP systems without attached PilT/VirB11-like family members[14], and PilB and PilT migrate dynamically in some bacteria while the core T4aP proteins are anchored in the cell envelope[44,45]. Thus, we propose that at physiological ATP and ADP concentrations, when it is not engaged with the T4aP, PilT preferentially adopts the CCCCCC conformation.

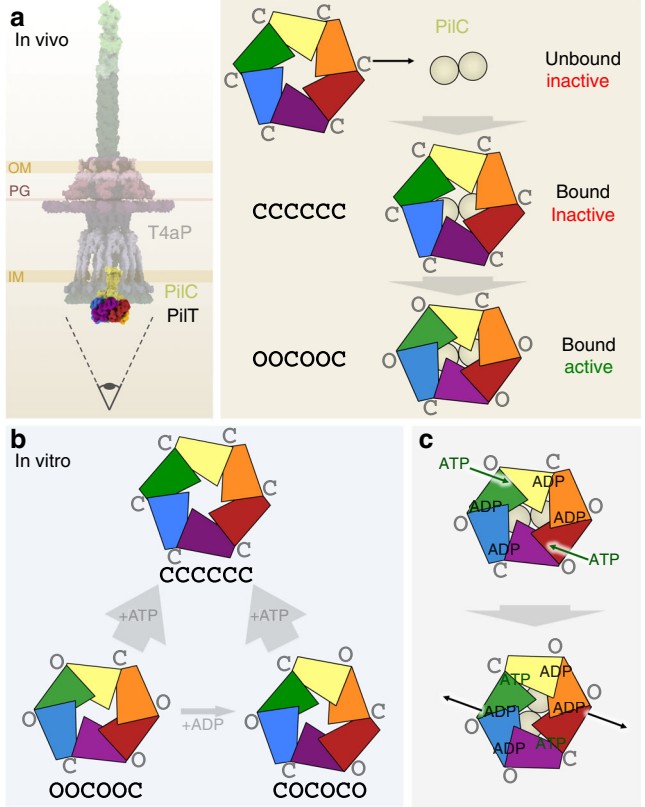

**Fig. 6** Proposed models of function. **a** In vivo model of PilT conformations. The entire T4aP is shown with an eye symbol (left) indicating the viewing direction for the model (right). This model suggests that the CCCCCC and OOCOOC conformations are of in vivo functional relevance. **b** In vitro model of PilT conformations. This model illustrates that in vitro PilT is a mixture of OOCOOC and OCOCOC conformations; added ADP can modestly increase the proportion of the OCOCOC conformation, while added ATP strongly favors the CCCCCC conformation. **c** Updated model of ATP binding and hydrolysis in PilT. ADP is predicted to be retained at two open interfaces, so only the other two open interfaces are available for binding ATP, promoting a single direction of rotation

Our mutational and coevolution analyses support the in vivo importance of the CCCCCC conformation. From BACTH data, the CCCCCC-targeting R123D PilT mutation impairs PilC interaction. Given this analysis supporting its importance, it was initially surprising that the closed-interface-destabilizing D31K mutation and the open-interface-stabilizing K58A mutation promote hypertwitching. These results suggest that the active form of PilT is likely to contain open interfaces. We propose that after binding to PilC in the CCCCCC conformation, PilT converts to an open-interface-containing conformation to power pilus depolymerization (Fig. 6a). Our data suggest that the only open-interface-containing PilT conformations found in solution are OOCOOC and OCOCOC. In the absence of a co-structure of PilT and PilC, the specific motor conformation of PilT remains unclear. PilC is proposed to be a dimer in vivo[15] and PilC-like proteins have crystallized with $C_2$ and asymmetric-pseudo-$C_2$ symmetry[46,47]; thus, when PilT binds PilC, we anticipate that the interaction would induce $C_2$ symmetry in PilT. Since the OOCOOC conformation of PilT is the only $C_2$ symmetric conformation of PilT found in solution, we propose that the active motor conformation is the OOCOOC conformation. Adopting other conformations in the presence of ATP may be a strategy to limit unnecessary hydrolysis in the absence of other T4aP

proteins, and could explain why the activity of PilT/VirB11-like family members is notoriously low in vitro[11,39].

In our previous model of OOCOOC PilT function, we suggested that binding of two ATPs to opposite open interfaces caused them to close[18]. This is consistent with our observations that bound ATP correlates with the closed interface. Closure of two interfaces was predicted to open the neighboring closed interfaces to enable release of ADP[18]. Our data herein suggest that ADP may not be released immediately. Unlike in PilB[Gm] structures[18], ADP was found in both the open and closed interfaces in PilT[Gm] structures. This affinity for ADP for the open interface is consistent with a model in which PilT temporarily retains ADP for an additional round of ATP hydrolysis after the closed interface opens (Fig. 6c). Such a mechanism would parallel that of PilB[18] despite its different patterns of open and closed interfaces. The consequence of this nuance would be that for PilB and PilT, only two open interfaces would be available at any one time for binding ATP. This scenario would commit PilT to a single direction of ATP binding and hydrolysis and thus a single direction of pore rotation. Consistent with the distinction between CCOCCO and OOCOOC conformations promoting PilB-like and PilT-like functions, respectively, we note that while PilB adopted the CCOCCO conformation in solution here and elsewhere[27], the equivalent $C_2$ symmetric conformation adopted by PilT[Gm] and PilT[Pa] is the OOCOOC conformation. Thus, the preferred conformations of PilB and PilT in solution correlate with function. Given that neither PilT[Gm] nor PilT[Pa] crystallized in the OOCOOC conformation, individual PilT/VirB11-like crystal structures should be interpreted with caution in the absence of accompanying cryoEM analysis.

In addition to the OOCOOC conformation, PilT is also found in the $C_3$ symmetric OCOCOC conformation in solution. The function of this conformation remains unclear. That a motor capable of rotating a substrate protein that would switch between $C_2$ and $C_3$ symmetries during its catalytic cycle is unprecedented. As judged by the similarity between open and closed interfaces between conformations, it may have been prohibitively difficult during evolution to stabilize the OOCOOC conformation without also stabilizing other conformations. Perhaps, evolution favored the relative stability of open versus closed interfaces rather than particular hexamer conformations. Thus, the relevant difference between a retraction and extension ATPase may be the relative stabilities of their open- and closed interfaces.

Although the CCOCCO PilT conformation was not observed in solution, our finding that a single PilT otholog can adopt both CCOCCO and OOCOOC conformations may be critical for understanding PilT/VirB11-like ATPase function and evolution. This finding suggests that PilT[Gm] and potentially other PilT/VirB11-like family members could have the capacity to switch between OOCOOC powered counterclockwise pore rotation (i.e., pilin depolymerization), and CCOCCO powered clockwise pore rotation (i.e., pilin polymerization), blurring the line between extension and retraction ATPases. Indeed, PilT is inexplicably essential for T4aP pilin polymerization in *Francisella tularensis*[48]. Similarly, some T4P-like systems — including the T2S, T4bP, T4cP pilus, and even some T4aP systems — have been shown to retract their filaments in the absence of a dedicated retraction ATPase or in PilT-deleted backgrounds[49–51]. A similar conformation switch could also explain how FlaI switches between clockwise and counterclockwise archaellum rotation[52]. Such a switch might easily be regulated by post-translational modifications or alternate partner-protein interactions that modulate the relative stability of open- versus closed interfaces. Indeed, evidence emerged during the completion of this paper that the single PilT/VirB11-like family member from the *Caulobacter* T4cP system powers both pilus polymerization and depolymerization[53].

It may be that the last common PilT/VirB11-like family member ancestor catalyzed both clockwise and counterclockwise rotation, facilitating both pilus polymerization and depolymerization, and only more recently have PilB and PilT specialized to perform separate functions.

## Methods

**Cloning and mutagenesis**. PilT4 from *G. metallireducens* GS-15 was PCR amplified from genomic DNA by using primers P168 and P169 (Supplementary Table 2), digested with NdeI and XhoI, and cloned into pET28a with a thrombin-cleavable hexahistidine tag to create pET28a:PilT$^{Gm}$. PilT from *P. aeruginosa* PAO1 was PCR amplified from genomic DNA by using primers P220 and P221, digested with KpnI and XbaI, and cloned into pBADGr for arabinose-inducible expression to create pBADGr:PilT$^{Pa}$. pKT25::PilT$^{Pa}$ and pBADGr:PilT$^{Pa}$ were mutated by site-directed mutagenesis (Quikchange II, Agilent) to generate E204A, H229A, R239A, E219K, D31K, E258A, R294E, D242K, T216R, R123D, K58A, K58Q, and K58R mutants by using the eponymously named primers (Supplementary Table 2). Plasmid sequences were verified by TCAG sequencing facilities (The Hospital for Sick Children, Canada).

**Expression and purification**. *E. coli* BL21-CodonPlus® cells (Strategene, Supplementary Table 1) were transformed with pET28a:PilT$^{Pa}$, pET28a:PilT$^{Pa}$, or pET28a:PilB$^{Gm}$ and grown in 4 L of lysogeny broth (LB) with 100 µg/ml kanamycin at 37 °C to an $A_{600}$ of 0.5–0.6, then protein expression was induced by the addition of isopropyl-D-1-thiogalactopyranoside (IPTG) to a final concentration of 1 mM, and the cells were grown for 16 h at 18 °C. Cells were pelleted by centrifugation at $9000 \times g$ for 15 min. Cell pellets were subsequently resuspended in 40 ml of binding buffer (50 mM Tris-HCl, pH 7, 150 mM NaCl, and 15 mM imidazole). Subsequent to crystallization of methylated OCOCOC PilT, the buffer was optimized to improve PilT$^{Gm}$ thermostability — the pH was increased, HEPES was used instead of Tris, the concentration of NaCl was increased, and glycerol was added; hereafter the binding buffer for PilT$^{Gm}$ was 50 mM HEPES, pH 8, 200 mM NaCl, 10% (v/v) glycerol, and 15 mM imidazole. For PilB$^{Gm}$, cell pellets were resuspended in 40 ml of 50 mM Tris, pH 7.5, 150 mM NaCl, and 50 mM imidazole. For PilT$^{Pa}$, cell pellets were resuspended in 40 ml of 100 mM Tris, pH 7.0, 150 mM NaCl, 30 mM citrate, 150 mM (NH$_4$)$_2$SO$_4$, 15% (v/v) glycerol, 5 mM MgSO$_4$, and 50 mM imidazole. After resuspension in binding buffer, the cells were lysed by passage through an Emulsiflex-c3 high-pressure homogenizer, and the cell debris removed by centrifugation for 45 min at $40000 \times g$. The resulting supernatant was passed over a column containing 5 ml of pre-equilibrated Ni-NTA agarose resin (Life Technologies, USA). The resin was washed with ten column volumes of binding buffer and eluted over a gradient of binding buffer to binding buffer plus 600 mM imidazole. Purified PilT$^{Gm}$ was additionally purified with a HP anion exchange column pre-equilibrated with binding buffer; the flow-through contained PilT$^{Gm}$. PilT$^{Gm}$, PilT$^{Pa}$, or PilB$^{Gm}$ was then further purified by size-exclusion chromatography on a HiLoad™ 16/600 Superdex™ 200-pg column pre-equilibrated with binding buffer without imidazole or glycerol. For the OCOCOC PilT structure with full-occupancy ADP, 2 mM ATP and 2 mM MgCl$_2$ were added just prior to size-exclusion chromatography. For the crystallization of methylated OCOCOC PilT, the size-exclusion chromatography buffer was 50 mM HEPES, pH 7, 150 mM NaCl, 10% v/v glycerol, and subsequent to purification PilT$^{Gm}$ was reductively methylated overnight (Reductive Alkylation Kit, Hampton Research), quenched with 100 mM Tris-HCl, pH 7, and the size-exclusion chromatography step was repeated. All purified proteins were used immediately.

**Crystallization, data collection, and structure solution**. For crystallization, purified PilT$^{Gm}$ was concentrated to 15 mg/ml (4 mg/ml for methylated OCOCOC PilT) at $3000 \times g$ in an ultrafiltration device (Millipore). Crystallization conditions were screened by using the complete MCSG suite (MCSG 1–4) (Microlytic, USA) using a Gryphon LCP robot (Art Robbins Instruments, USA). Crystal conditions were screened and optimized using vapor diffusion at 20 °C with Art Robbins Instruments Intelli-Plates 96-2 Shallow Well (Hampton Research, USA) with 1 µl of protein and 1 µl of reservoir solution. For the methylated OCOCOC PilT structure, the reservoir solution was 200 mM ammonium sulfate, 100 mM Bis-Tris-HCl, pH 6.5, and 25% (w/v) PEG3350. For cryoprotection of the methylated OCOCOC PilT crystal, 2 µl of 50% (w/v) ethylene glycol, and 50% (v/v) reservoir solution was added to the drop containing the crystal for 10 s prior to vitrification in liquid nitrogen. For the ADP-bound OCOCOC PilT structures, the reservoir solution was 11.5% (w/v) PEG3350, 200 mM L-proline, and 100 mM HEPES — pH 7.1 for the partial-occupancy ADP structure or pH 7.6 for the full-occupancy ADP structure. For cryoprotection of ADP-bound OCOCOC PilT crystals, 2 µl of 7.5% (w/v) xylitol, 15% (w/v) sucrose, and 50% (v/v) reservoir solution was added to the drop containing the crystal for 10 s prior to vitrification in liquid nitrogen. For the CCCCCC PilT structure, the protein solution also contained 2 mM ATP and 2 mM MgCl$_2$. The reservoir solution was 9% (w/v) PEG4000, 100 mM MES, pH 6, and 200 mM MgCl$_2$. For cryoprotection of the CCCCCC PilT crystal, 2 µl of 50% (w/v) ethylene glycol, and 50% (v/v) reservoir solution was added to the drop containing the crystal for 10 s prior to vitrification in liquid nitrogen. For the CCOCCO PilT

structure, the protein solution also contained 2 mM ANP and 2 mM MgCl$_2$. The reservoir solution was 2% (w/v) benzamidine-HCl, 100 mM HEPES-HCl, pH 7.9, 200 mM ammonium acetate, and 17.5% (w/v) PEG3350. For cryoprotection of the CCOCCO PilT crystal, 2 µl of 15% (w/v) xylitol, and 50% (v/v) reservoir solution was added to the drop containing the crystal for 10 s prior to vitrification in liquid nitrogen.

Diffraction data were collected by using synchrotron X-ray radiation as noted in Supplementary Table 3. The data were indexed, scaled, and truncated by using XDS[54]. The CCOCCO PilT data were anisotropically truncated and scaled by using the Diffraction Anisotropy Server[55]. PHENIX-MR[56] was used to solve the structures of PilT$^{Gm}$ by molecular replacement with PDB 3JVV preprocessed by the program Chainsaw[57]. In every case, the resulting electron density map was of high enough quality to enable building the PilT protein manually in COOT[58]. Through iterative rounds of building/remodeling in COOT[58] and refinement in PHENIX-refine[59] the structures were built and refined. Rosetta refinement in PHENIX[60] helped improve models early in the refinement process, while refinement in the PDB-redo webserver[61] helped improve models late in the refinement process. CCCCCC PilT was refined with individual B-factors, all other structures were refined with a single B-factor per residue. The occupancy of the nucleotides in the partial-occupancy-ADP OCOCOC PilT structure was estimated by PHENIX-refine, restricting all atoms in a nucleotide to a uniform occupancy. Progress of the refinement in all cases was monitored by using $R_{free}$.

**Enzyme-coupled ATPase assay**. Enzyme-coupled ATPase assays were performed as done elsewhere[62], with minor modifications. Briefly, the reaction buffer included 40 U/ml lactate dehydrogenase (Sigma), 100 U/ml pyruvate kinase (Sigma), 100 mM NaCl, 2 mM MgCl$_2$, 25 mM KCl, 0.8 mM nicotinamide adenine dinucleotide, 10 mM phosphoenolpyruvate, 5 mM ATP, and 0.088 mM PilT$^{Gm}$. The pH of the reaction buffer was set with a 200 mM MES and 200 mM HEPES dual buffer, at pH 5.5, 6.0, 6.5, 7.0, 8.5, or 9.0. The reaction volume was 100 µl. Conversion of NADH to NAD$^+$ — proportional to the ADP produced by the hydrolysis of ATP — was monitored by measuring the $A^{340}$ every 2 min at 25 °C for 2 h. Initial reaction rates were used. Control experiments were performed (without added PilT$^{Gm}$) spiking the reaction buffer at different pH values with 2 mM ADP; conversion of all NADH to NAD$^+$ occurred almost immediately at every pH value used herein indicating that the reagents were not rate limiting.

**Identifying hexamer symmetry**. Each hexamer (extracted from the corresponding PDB coordinates) was aligned in the PyMOL Molecular Graphics System, version 2.2 (Schrodinger, LLC 2010) against the same hexamer six times. Specifically, all six chains were aligned with the $n + 1$ chains, $n + 2$ chains, $n + 3$ chains, $n + 4$ chains, $n + 5$ chains, or $n + 6$ chains, and the RMSD$^{C\alpha}$ of the alignments was noted. If the RMSD$^{C\alpha}$ was below 1 Å, the rotation was considered to be equivalent, and if the RMSD$^{C\alpha}$ was above 4 Å the rotation was considered to be distinct; this was used to define the symmetry of the hexamer. For example, in the CCOCCO PilT structure, the RMSD$^{C\alpha}$, the $n + 3$, and $n + 6$ alignments were equivalent, while the $n + 1$, $n + 2$, $n + 4$, and $n + 5$ alignments were distinct, consistent with $C^2$ rotational symmetry. If the RMSD$^{C\alpha}$ was between 1 and 4 Å, the rotation was considered to be pseudosymmetric. For example, in the ADP-bound OCOCOC PilT structures, the $n + 1$, $n + 3$, and $n + 5$ alignments were distinct, while the RMSD$^{C\alpha}$ of the $n + 2$, $n + 4$, and $n + 6$ alignments was ~3.5 Å — a pattern consistent with pseudo-$C_3$ symmetry.

**Plotting the residues at the packing-unit interface**. CMview[63] was used to identify residues that contact one another. The contact type was set to 'ALL' (i.e., every atom available in the structure) and the distance cutoff was set to 4 Å. To identify residues that contact one another even in low-resolution structures, where side-chain modeling is less definitive, additional residue contacts were identified by setting the contact type to 'BB' (i.e., backbone atoms) and the distance cutoff to 8 Å. To identify tertiary structure contacts, the N2D (residues 1–100 in PilT$^{Gm}$) and CTD (residues 101–353 in PilT$^{Gm}$) were loaded separately — to simplify analysis, the linker between the N2D and CTD was considered to be part of the CTD. To identify packing-unit-forming contacts, the N2D$^n$ and CTD$^{n+1}$ from adjacent chains were loaded together, and the previously identified tertiary structure contacts were subtracted from this contact list. To identify contacts that form the interface between two adjacent packing units, two adjacent packing units were loaded together, and then the previously identified tertiary structure and packing-unit-forming contacts were subtracted from this contact list. For clusters of residues that are in proximity, only the most prominent contacts (salt bridges and dipolar interactions) were considered.

**Identifying open versus closed packing-unit interfaces**. The open- and closed interfaces of PilT$^{Gm}$ crystal structures were initially identified by qualitative comparison with the characterized open- and closed interfaces of PilB$^{Gm}$. Subsequent to our finding that the open and closed interfaces are correlated with intermolecular T221–H230 and T221–L269 distances, the α-carbon distances between these residues were measured in PilT$^{Gm}$. In other PilT/VirB11-like family members, the α-carbon distances between the residues that correspond with PilT$^{Gm}$ residues T221, H230, or L269 were used. If the T221–H230 distance was

greater than 11 Å and the T221–L269 distance was less than 11 Å, this interface was classified as O. If the T221–H230 distance was less than 9 Å and the T221–L269 distance was more than 12 Å, this interface was classified as C. Interfaces that did not meet these criteria were classified as X-interfaces.

**Identifying evolutionarily coupled residues**. The PilT[Pa] amino acid sequence was analyzed by using the *EVcouplings* webserver[34] with default parameters. Homologs were identified, aligned, and analyzed — 30,421 in total. This analysis did not include the last 60 C-terminal residues of PilT as the alignment had too many gaps in this region and default parameters enforce 30% maximum gaps allowed. Relaxing this parameter to 50 and 75% maximum gaps allowed more C-terminal residues to be included in the analysis, though some of the evolutionarily coupled residue pairs identified with default parameters were not discovered. To compensate, we merged the evolutionarily coupled residues identified with default parameters, with 50% maximum gaps, and 75% maximum gaps. This analysis yielded overall coverage from residue 18 to 347, though coupled residues in the last 60 C-terminal residues likely have a lower likelihood of being identified. Only residue pairs with a PLM score greater than 0.2 were included in subsequent analysis. To better understand the significance of these evolutionarily coupled residues, they were compared with the tertiary structure contacts, packing-unit contacts, and open- and closed-interface contacts identified in CMview.

**CryoEM analysis**. Newly purified PilT[Gm] at 0.5 mg/ml in binding buffer without imidazole or glycerol was incubated with and without 1 mM $MgCl_2$ plus 1 mM ATP, 1 mM $MgCl_2$ plus 0.1 mM ADP, 1 mM $MgCl_2$ plus 1 mM ADP, 1 mM $MgCl_2$ plus 1 mM ATP and 0.1 mM ADP, or 1 mM $MgCl_2$ plus 1 mM ATP and 1 mM ADP at 4 °C for 10 min before preparing cryoEM grids. PilT[Pa] at 0.75 mg/ml or PilB[Gm] at 0.6 mg/ml in binding buffer without imidazole or glycerol were also incubated without added nucleotide at 4 °C for 10 min before preparing cryoEM grids. Three microliters of protein sample was applied to nanofabricated holey gold grids[64–66], with a hole size of ~1 μm and blotted by using a modified FEI Vitribot Mark III at 100% humidity and 4 °C for 5.5 s before plunge freezing in a 1:1 mixture of liquid ethane and liquid propane held at liquid nitrogen temperature[67].

CryoEM data was collected at the Toronto High-Resolution High-Throughput cryoEM facility. Micrographs from untilted specimens were acquired as movies with a FEI Tecnai F20 electron microscope operating at 200 kV and equipped with a Gatan K2 Summit direct detector camera. Movies, consisting of 30 frames at two frames per second, were collected with defocus values ranging from 1.2 to 3.0 μm. Data were recorded with an exposure rate of 5 electrons/pixel/s with a calibrated pixel size of 1.45 Å/pixel. For the 40° tilted data collection, micrographs were acquired as movies with a FEI Titan Krios electron microscope (Thermo Fisher Scientific) operating at 300 kV and equipped with a Falcon 3EC direct detector camera. Movies, consisting of 30 frames at 2 s per frame, were collected with defocus values ranging from 1.7 to 2.5 μm. Data were recorded with an exposure rate of 0.8 electrons/pixel/s with a calibrated pixel size of 1.06 Å/pixel.

All image processing of the cryoEM data was performed in cryoSPARC v2.8.0 (ref.[36]) (Supplementary Table 4). Movie frames were aligned with an implementation of alignframes_lmbfgs within cryoSPARC v2 (ref.[68]) and CTF parameters were estimated from the average of aligned frames with CTFFIND4 (ref.[69]). Initial 2D class averages were generated with manually selected particles; these classes were then used to select particles. Particle images were selected and beam-induced motion of individual particles corrected with an improved implementation of alignparts_lmbfgs within cryoSPARC v2 (ref.[68]). For the 40° tilted data, an implementation of *GCTF*[37] wrapped within cryoSPARC v2 was used to refine the micrograph CTF parameters while also locally refining the defocus for individual particles with default parameters (local_radius of 1024, local_avetype set to Gaussian, local_boxsize of 512, local_overlap of 0.5, local_resL of 15, local_resH of 5, and refine_local_astm set to Z-height). Particle images were extracted in 256 × 256-pixel boxes. Candidate particle images were then subjected to 2D classification. For the particles that preferentially adopted top views, comparison of 2D class averages of these top views with 2D projections of PilT structures was used to identify the corresponding conformation. 2D projections of PilT structures were generated by using genproj_fspace_v1_01 (J. Rubinstein, https://sites.google.com/site/rubinsteingroup/3-d-fourier-space). For the purposes of 3D classification, particle images contributing to 2D classes without high-resolution features were removed. For samples with tilted views, ab initio reconstruction was performed by using two to four classes. Ab initio classes consistent with hexamers were used as initial models for heterogeneous refinement; particles from 3D classes that did not converge at this stage were removed. Particles from distinct 3D classes were then subjected to homogeneous refinement.

Molecular models could be built into these maps by fitting rigid packing units of the PilT[Gm] crystal structures into the maps in Chimera[70]. These models were refined against the maps in Phenix-Refine[71] with reference model restraints to the ATP-bound PilT[Gm] crystal structure (or for the PilB[Gm] model, PDB 5TSH), enabling the following refinement options: minimization_global, rigid_body, simulated_annealing, and adp. The overall quality of the maps was impacted by the anisotropy from preferred orientations, so side chains were not modeled.

**In vivo transcomplementation assays**. *P. aeruginosa* PAO1 *pilT*::FRT (Supplementary Table 2) was electroporated with pBADGr, pBADGr::PilT[Pa], or pBADGr::PilT[Pa] derivative mutant constructs for transcomplementation of PilT. Twitching assays were performed in 150 mm by 15-mm petri polystyrene dishes (Fisher Scientific) with 30 μg/ml gentamicin for 18 h at 37 °C[72,73]. After this incubation, the agar was then carefully discarded and the adherent bacteria were stained with 1% (w/v) crystal violet dye, followed by washing with deionized water to remove unbound dye. Twitching zone areas were measured by using ImageJ software[74]. Twitching motility assays were performed in six replicates.

Surface pili were analyzed as previously described[75], with the exception that sheared supernatant pili and flagellins were precipitated with 100 mM $MgCl_2$ incubated at room temperature for 2 h prior to pelleting. These pellets were resuspended in 100 μl of 1× SDS-PAGE sample buffer, 10 μl of which was loaded onto a 20% SDS-PAGE gel. In parallel, western blot analysis was performed on the cells used to produce the pili to confirm mutant stability by using rabbit polyclonal anti-PilT antibodies (Supplementary Table 2).

In preparation for the PO4 phage infection assay, 25 ml of LB with 1.5% (w/v) agar was solidified in 150 mm by 15-mm polystyrene petri dishes (Fisher Scientific). On top of this layer, 8 ml of 0.6% (w/v) agar preinoculated with 1 ml of $A_{600} = 0.6$ transcomplemented *P. aeruginosa* PAO1 *pilT*::FRT was poured and solidified. Three microliters of $10^4$ plaque-forming units per ml PO4 phage were then spotted onto these plates in triplicate and incubated at 30 °C for 16 h before images of the plates were acquired.

**Bacterial two-hybrid analysis (BACTH)**. BACTH analysis was performed as done previously[38] with minor adjustments. Briefly, *E. coli* BTH101 cells (Supplementary Table 2) were co-transformed with pUT18C::pilC, or pUT18C::pilT and mutants of pKT25::pilT. Preliminary tests suggested that prolonged incubation in induction conditions at 25 °C improved the signal to noise, so three colonies from each of these transformations were individually streaked onto MacConkey-Maltose Agar plus 100 μM ampicillin, 50 μM kanamycin, and 0.5 mM IPTG and incubated overnight at 25 °C. A single colony from these plates were then used to inoculate 500 μl of LB plus 100 μM ampicillin, 50 μM kanamycin, and 0.5 mM IPTG at 25 °C until the $A_{600}$ was 1.0. Five microliters of this solution was then used to inoculate 500 μl of LB plus 100 μM ampicillin, 50 μM kanamycin, and 0.5 mM IPTG at 25 °C for 16 h. Thirty-five microliters of this was then used to inoculate 600 μl of LB plus 100 μM ampicillin, 50 μM kanamycin, and 0.5 mM IPTG at 25 °C for 1 h, then at 18 °C until the $A_{600} = 0.6$. All the replicates were then normalized to 600 μl and $A_{600} = 0.6$, then pelleted at 3200 × *g* for 5 min. The supernatant was carefully removed, and the pellet was resuspended in 100 μl of 200 mM $Na_2HPO_4$, pH 7.4, 20 mM KCl, 2 mM $MgCl_2$, 0.8 mg/ml cetyltrimethylammonium bromide detergent, 0.4 mg/ml sodium deoxycholate, and 0.54% (v/v) β-mercaptoethanol. After 5 min of incubation at room temperature, 10 μl of this solution was then transferred to a 96-well clear-bottom plate. One hundred and fifty microliters of a second solution was then added: 60 mM $Na_2HPO_4$ pH 7.4, 40 mM KCl, 20 μg/ml cetyltrimethylammonium bromide detergent, 10 μg/ml sodium deoxycholate, 0.27 % (v/v) β-mercaptoethanol, and 1 mM *ortho*-nitrophenyl-β-galactoside. The $A_{420}$ and $A_{550}$ were measured every 2 min for 30 min at 30 °C. β-galactosidase activity in Miller units was calculated by finding the slope of $1000 \times (A_{420} - 1.75 \times A_{550})/(0.6$ absorbance units $\times 0.06$ ml) over time (minutes) of the linear portion of the initial reaction. BACTH assays were performed in triplicate.

**Reporting summary**. Further information on research design is available in the Nature Research Reporting Summary linked to this article.

## Data availability

Structural data that support the findings of this study have been deposited in the Protein Data Bank with the accession codes 6OJY, 6OJZ, 6OK2, 6OJX, 6OKV, 6OLL, 6OLK, 6OLM, and 6OLJ, as well as the Electron Microscopy Data Bank with the accession codes EMD-20116, EMD-20115, EMD-20117, and EMD-20114. The source data underlying Figs. 1i, 4a, b and 5a, b and Supplementary Fig. 2 are provided as a Source Data file. All other data are available within the paper and its Supplementary information files or are available from the corresponding author upon reasonable request.

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

## Acknowledgements

We thank Roland Pfoh, Jeff Lee, and Natalie C. Bamford, as well as Zev Ripstein, Hui Guo, and Thamiya Vasanthakumar for technical assistance during crystal diffraction screening/data collection and cryoEM grid freezing/data collection, respectively. Shane Caldwell is thanked for technical assistance and helpful discussions. This work was supported by a grant from the Canadian Institutes for Health Research (CIHR) MOP 93585 to L.L.B. and P.L.H. M.M. was supported by a CIHR Doctoral Studentship and Ontario Graduate Scholarship during these studies. P.L.H. and J.L.R. are recipients of Canada Research Chairs. Funds for the X-ray and EM facilities at The Hospital for Sick Children were provided in part by the Canadian Foundation for Innovation and the Government of Ontario. Protein crystals were grown and screened in the Structural & Biophysical Core (SBC) Facility at The Hospital for Sick Children. The crystallographic and graphics programs used in this study were accessed by using SBGrid.

## Author contributions

All authors edited the final paper. M.M., S.T., J.L.R., L.L.B. and P.L.H. were involved with project design. S.N. and M.M. generated the PilT mutants. S.B. and M.M. collected tilted cryoEM movies. All other experimental work was completed by M.M.

## Competing interests

The authors declare no competing interests.
