## [Peer Review File · Nature Communications]

Reviewers' comments:

Reviewer #1 (Remarks to the Author):

Before the specific points, I have two general comments to make on this detailed manuscript. First, there are major challenges inherent in attempting to infer a mechanism of energy transduction (how ATP hydrolysis is converted into the mechanical work required for pilus assembly) from structural studies carried out *in vitro* on isolated ATPases. In the absence of interacting partner proteins and regulatory components, as would be the case *in vivo*, deductions about mechanism are always going to be speculative. The manuscript makes a valiant attempt at a comprehensive assessment of existing structural information on this family of ATPases but it (mostly) still suffers from this drawback. Second, there is already a substantial literature on the structures of these ATPases which has attempted to do this.

Specific points

1. Range of the PilT-like family

The Introduction discusses the background to the role of ATPases in type IV pilus biogenesis but the analysis in Fig 3, and accompanying text, includes ATPases (described as 'PilT-like') which are involved in other secretion systems eg GspE (T2SS); VirB11 (T4SS). It could well be that this subfamily of AAA+ ATPases share a common mechanism; however I think the text in the relevant parts needs to acknowledge that the network of interacting proteins to which these proteins are linked will be different in some cases. A clearer definition of what the authors mean by 'PilT-like' (in the Introduction) would also help the reader understand the scope of the study.

2. GSPII domains

The PilB-like ATPases harbor GSPII domains at their N-termini; if the full length PilB(Gm) was used, why were no additional domains detected in the cryoEM density (Fig 4k)? Perhaps they were removed by proteolysis during sample preparation? The presence of additional domains marks the PilB ATPases out as different from their PilT counterparts (this observation is related to point #1).

3. 2D classification

For the cryoEM analysis, it is clear that the PilT and PilB hexamers preferentially adopt face-on rather than side-on views. Tilt data was only collected for PilTGm in one state (Table S2) and it seems from Fig S3 that side views are rare, probably leading to anisotropy in resolution. Qualitatively, from the appearance of the maps in Fig S3, the resolutions claimed seem optimistic- at 4-5 Angstroms resolution helices in cryo maps start to look spiral and lumpy rather than tubular. It would be useful to include some further detail in Supplementary Material- some Euler angle distributions and raw data examples. In addition, the details of individual projections in Fig S2 are not clear at this digital resolution. It would help to enlarge these images and distribute them over several figures.

This latter point is important because a central argument throughout the manuscript is that the authors claim that they can distinguish different conformational states from 2D projections, or 3D reconstructions which are dominated by face-on views. For example, in lines 378-379, the authors state that they were able to distinguish (and, critically, quantify) C3OCOCOC from C2OOCOOC, but how reliable are these classifications and the associated quantification? The majority of 2D class averages in Fig S2 are marked as being '...too blurry to identify the conformation'. Does this mean that a high proportion of 2D class averages were omitted from such calculations because they cannot be classified? If that is the case, it suggests that the assignment to precise proportions of different conformational states (eg C3OCOCOC/C2OOCOOC) is not robust. More detail is needed to justify the quantification of different conformational states, particularly from 2D projection data.

4. Site-directed mutagenesis

I was less convinced by the conclusions drawn from the mutagenesis experiments (lines 483-552). The rationale, as I understand it, is to disrupt interactions which stabilize the O or C conformations, and then examine twitching motility, pilus formation and other phenotypes. One problem is that in only one case (K58A) is the consequence of the mutation on the distribution of conformational states actually demonstrated- for the other mutations the effect of the selected mutations is inferred. Even if this could be demonstrated *in vitro*, other explanations for the

phenotypes observed are possible (eg indirect effects on ATPase activity). Some mutations lead to weak PilT expression (not necessarily to be equated with stability-line 501); others cause low levels of sheared pilin, even with low twitching motility (E258A). For the PilC binding, where an effect on PilC binding is observed, can a direct role for these residues in binding PilC be ruled out? It should also be noted that the bacterial 2 hybrid may not necessarily reproduce the in vivo PilT-PilC interaction, where it could be affected by the presence of other pilus biogenesis components. In general, interpretation of these results is difficult because our current knowledge of the structural details of the PilT-PilC interaction and the ATPase reaction cycle is rudimentary.

Reviewer #2 (Remarks to the Author):

The type IV pili machine has important functions in motility, biofilm formation and virulence. It stimulates the extension and retraction of type IV pili in a large number of bacteria. Generally, these two processes are stimulated by the ATPases PilB for extending a pilus and PilT for retracting a pilus. Over the years, several structures of these and homologous ATPases have been determined. The structures suggest that these ATPases function as hexamers that undergo large conformational changes during ATP binding and hydrolysis. Moreover, it has previously been suggested that PilB and PilT rotate the inner membrane protein PilC in opposite directions to stimulate the incorporation of pilin subunits at the base of the growing pilus or to remove pilin subunit from the base of the shrinking pilus during retractions. However, the details of the interaction to PilC and how PilB/PilT would rotate PilC are not well-understood. Here, the authors solve the structure of PilT from *Geobacter metallireducens* using crystallography and single particle EM. Different conformations are observed. These structural analyzes were followed up by mutagenesis of residues at identified interfaces using PilT in *P. aeruginosa* and phenotypic characterization. Effects on PilT function is observed as a consequence of these substitutions.

Semantically, I am not sure that referring to the various ATPase discussed as motor PilT-like ATPases is helpful. After all, PilT is the least widespread of the PilB, GspE and PilT ATPases. From sequence alignments, it is also clear that these three ATPases are very different. For a non-structural biologist the manuscript is incredibly difficult to read and very condense. Also, in the figures it is really difficult to follow and appreciate the differences that the authors are discussing. So (again from a non-structural biologist's point of view), it does not really become clear how the new PilT structures are different from previous structures and which new lessons have been learned. The in vivo analyzes in *P. aeruginosa* are straight-forward and, as expected, most substitutions at the subunit interface interfere with function. Also here, it would be incredibly helpful if the authors would label the residues in Figure 5 in such a way that it is clear in which conformation they expect the PilT variants to be locked in.

Reviewer #3 (Remarks to the Author):

Summary:

This article by McCallum et al., provides a detailed structural analysis of PilT-like ATPases. These motors are essential to the function of type IV pili, which in turn are critical for the virulence of many bacterial pathogens. The authors contribute four new crystal structures and provide a thorough cryo-EM analysis leading to four further maps and models of PilT/PilB ATPases. The authors tie together the structural work presented in this manuscript with previous work and propose a comprehensive model of PilT function. Furthermore, coevolution analysis and PilT mutants provide functional validation for the conclusions drawn. The manuscript is clear and well written and provides novel insights to the current state of knowledge of PilT ATPases. Experiments are well executed and appropriate statistical analyses are provided. This work is a valuable addition to the field and I, therefore, support publication with some minor modifications.

Minor Points:

1. Typo in legend to Figure 1: In the last line, the brackets around I in " $I | \sigma(I)$ " are missing (Page 4, line 131).

2. Question, Page 10: the particle distribution ratio between C3OCOCOC and C2OOCOOC between PiITGm and PiITPa is significantly different in the cryo-EM analysis. Do the authors have a suggestion as to the functional implication of this observation? Or do the authors believe that the C3OCOCOC conformation is not relevant in vivo at all?

3. Page 10, line 410: The authors state that no density consistent with nucleotide was present in the maps and that this is presumably due to the low resolution of the maps (4.0 and 4.1 Å). However in the 4.4 Å structure of PiITGm briefly incubated with ATP, the nucleotide density is visible. Therefore, I would be more inclined to attribute the lack of nucleotide density in the PiITGm without nucleotide structure to the fact that no nucleotide was added to begin with and/or that nucleotide that was potentially carried over from the E. coli expression system is present at low occupancy. Perhaps this sentence can be modified accordingly.

4. Page 20, cryo-EM analysis section: Was the microscope equipped with an energy filter? If so, what was the slit width? What was the total electron dose in $e/\text{Å}^2$?

5. Page 20, line 873: Perhaps reword defucuses to defoci or defocus range or values.

6. Page 21, line 914: typo. "...adherent bacteria were stained..."

7. Page 27, line 993: typo? Should it read "...unit is composed of the N2Dn and CTDn+1..."

Response to reviewers' comments:

REVIEWER 1

Before the specific points, I have two general comments to make on this detailed manuscript. First, there are major challenges inherent in attempting to infer a mechanism of energy transduction (how ATP hydrolysis is converted into the mechanical work required for pilus assembly) from structural studies carried out in vitro on isolated ATPases. In the absence of interacting partner proteins and regulatory components, as would be the case in vivo, deductions about mechanism are always going to be speculative. The manuscript makes a valiant attempt at a comprehensive assessment of existing structural information on this family of ATPases but it (mostly) still suffers from this drawback.

RESPONSE: We appreciate this reviewer's comment and this is why our deductions are limited to the available data and possible alternatives. We have added the following text to emphasize the drawback of studying isolated components:

Line 642: "In the absence of a co-structure of PilT and PilC the specific motor conformation of PilT remains unclear."

Second, there is already a substantial literature on the structures of these ATPases which has attempted to do this.

RESPONSE: This study is the first attempt to unify the structural descriptions and analyses across the entire family of PilT/VirB11-like ATPases using the concept of inter-subunit contacts and their role in function. The key to our analysis is in our robust and quantitative definitions of open (O-) and closed (C-) interfaces that holds true across this diverse family of ATPases. Previous analyses that have focused on descriptions of individual chains do not robustly describe or adequately distinguish O- and C-interfaces. To emphasize this point, we included the following text in the discussion:

Line 592: "This is the first study to unify the structural description and analyses across PilT/VirB11-like family members."

Specific points

1. Range of the PilT-like family

The Introduction discusses the background to the role of ATPases in type IV pilus biogenesis but the analysis in Fig 3, and accompanying text, includes ATPases (described as 'PilT-like') which are involved in other secretion systems eg GspE (T2SS); VirB11 (T4SS). It could well be that this subfamily of AAA+ ATPases share a common mechanism; however I think the text in the relevant parts needs to acknowledge that the network of interacting proteins to which these proteins are linked will be different in some cases. A clearer definition of what the authors mean by 'PilT-like' (in the Introduction) would also help the reader understand the scope of the study.

RESPONSE: In the past, some authors have referred to this family as the PilT/VirB11-like family, which may be more semantically clear. Thus, we have replaced 'PilT-like ATPases' with 'PilT/VirB11-like family members' where relevant throughout the text. PilT/VirB11-like family members are not AAA+ ATPases (Iyer, L.M. *et al.* 2004. *Nucleic Acids Res.* 32(17):5260-5279). The majority of PilT/VirB11-like family members are involved in T4P-like systems and are thought to interact with PilC-like inner membrane platform proteins to assemble and/or disassemble a helical pilus. Thus, it is reasonable to assume that PilT/VirB11-like family members of T4P-like systems operate with a common mechanism. The T4SS lacks

a PilC-like inner membrane platform protein, so it is not clear if VirB11 has the same mechanism. This point has been made clearer in the text.

Line 588: “Based on our ability to clearly categorize all PilT/VirB11-like family members, we predict that PilT/VirB11-like family members of T4P-like systems operate with a common mechanism. The T4SS lacks a PilC-like inner membrane platform protein, so VirB11 or DotB may have distinct mechanisms.”

2. GSPII domains

The PilB-like ATPases harbor GSPII domains at their N-termini; if the full length PilB(Gm) was used, why were no additional domains detected in the cryoEM density (Fig 4k)? Perhaps they were removed by proteolysis during sample preparation?

RESPONSE: The reviewer is correct, even though the N-terminal His-tag and by extension the N-terminal domain was present during affinity purification, N-terminal domain is missing in the PilB cryoEM map. To address this point, we performed a mass spectrometry analysis of purified PilB used for cryoEM analysis and a summary of the data have been included in the source data file. The mass spectrometry data indicate that the N-terminal domain is still present. Given this finding, the most likely explanation for the lack of density for the N-terminal domain is that this domain is disordered relative to the core motor domains in the absence of its specific partner proteins. The following has been added to the text:
Line 455: “It should be noted that the N-terminal domain of PilB^{Gm}, known as the N1D, MshEN, or GSPII domain, was not observed in the 3D map although its presence was confirmed by trypsin digest followed by mass spectrometry (98% coverage from the His-tag to the C-terminus). It may be that in the absence of its binding partners, this domain is disordered relative to the core motor domains of PilB^{Gm}.”

The presence of additional domains marks the PilB ATPases out as different from their PilT counterparts (this observation is related to point #1).

RESPONSE: The presence of additional domains certainly differentiates PilT and PilB, but we believe that motor domain conformational differences are likely the more important structural difference in term of the mechanism of pilus assembly and disassembly. For example, FlaI, TadA, and T4bP assembly ATPases can assemble a pilus but do not have GSPII domain. If the GSPII domain were absolutely essential for pilus assembly it would have been conserved across all T4P-like system assembly ATPases during evolution. Only the core motor domains are conserved across all PilT/VirB11-like family members. It seems more plausible that the GSPII domain is important for PilB docking to the T4aP, and not necessarily the motor mechanism – which would be consistent with the GSPII domains being disordered relative to the core motor domains.

3. 2D classification

For the cryoEM analysis, it is clear that the PilT and PilB hexamers preferentially adopt face-on rather than side-on views. Tilt data was only collected for PilTGm in one state (Table S2) and it seems from Fig S3 that side views are rare, probably leading to anisotropy in resolution. Qualitatively, from the appearance of the maps in Fig S3, the resolutions claimed seem optimistic- at 4-5 Angstroms resolution helices in cryo maps start to look spiral and lumpy rather than tubular. It would be useful to include some further detail in Supplementary Material- some Euler angle distributions and raw data examples.

RESPONSE: We would like to draw the reviewer’s attention to Figure 4, which already has examples of the map quality, and Supplementary Figure 3 that already includes the Euler angle distributions (i.e. “particle distributions”). Some side chains could be observed (see maps in Figure 4) consistent with 4 Å

resolution. That being said, the overall quality of the maps was not as good as typical 4 Å resolution maps due to resolution anisotropy. Nonetheless, the conclusions made from the data are consistent with map quality. For example, side chains were not built into the models based on these maps. A note about this has been added to the text.

Line 938: “The overall quality of the maps was impacted by the anisotropy from preferred orientations, so side-chains were not modeled.”

In addition, the details of individual projections in Fig S2 are not clear at this digital resolution. It would help to enlarge these images and distribute them over several figures.

RESPONSE: We have increased, as requested, the resolution of Supplementary Figure 2. Higher resolution individual files have also been added to the source data.

This latter point is important because a central argument throughout the manuscript is that the authors claim that they can distinguish different conformational states from 2D projections, or 3D reconstructions which are dominated by face-on views. For example, in lines 378-379, the authors state that they were able to distinguish (and, critically, quantify) C3OCOCOC from C2OOCOOC, but how reliable are these classifications and the associated quantification?

RESPONSE: The proportion of particles classified as C_3^{OCOCOC} or C_2^{OOCOOC} in the 2D class averaging and the 3D heterogeneous refinement was within 3% (45:55 vs 48:52, respectively), consistent with these 2D classifications being reliable. Note that 3D classification was done autonomously and was independent of the 2D classification. The following was added to the text:

Line 380: “The particle distribution ratio was 48:52 between C_3 and C_2 maps, respectively; similar to that obtained by 2D classification, validating the 2D conformation assignments.”

The majority of 2D class averages in Fig S2 are marked as being ‘..too blurry to identify the conformation’. Does this mean that a high proportion of 2D class averages were omitted from such calculations because they cannot be classified? If that is the case, it suggests that the assignment to precise proportions of different conformational states (eg C3OCOCOC/C2OOCOOC) is not robust. More detail is needed to justify the quantification of different conformational states, particularly from 2D projection data.

RESPONSE: As with most cryoEM data, during data processing autopick selects a high proportion of particles that may be in thick ice or are just noise that subsequently do not classify easily and are blurry. Given that this is the first iteration of 2D classification (shown to illustrate that we did not omit inconvenient classes) it is not surprising that the majority of 2D classes are blurry. As mentioned above, the proportion of particles classified as C_3^{OCOCOC} or C_2^{OOCOOC} in the 2D class averaging and the 3D heterogeneous refinement was within 3% (45:55 vs 48:52, respectively), consistent with these 2D classifications being reliable. As stated above the 3D classification was done autonomously and was independent of the 2D classification.

4. Site-directed mutagenesis

I was less convinced by the conclusions drawn from the mutagenesis experiments (lines 483-552). The rationale, as I understand it, is to disrupt interactions which stabilize the O or C conformations, and then examine twitching motility, pilus formation and other phenotypes. One problem is that in only one case (K58A) is the consequence of the mutation on the distribution of conformational states actually demonstrated- for the other mutations the effect of the selected mutations is inferred. Even if

this could be demonstrated in vitro, other explanations for the phenotypes observed are possible (eg indirect effects on ATPase activity). Some mutations lead to weak PilT expression (not necessarily to be equated with stability-line 501); others cause low levels of sheared pilin, even with low twitching motility (E258A).

RESPONSE: As with any *in vivo* study, there are several plausible explanations for the observed phenotypes. We have added the following to the text to ensure the reader is aware of these alternative explanations.

Line 478: "The E219K and D31K mutants had reduced stability or expression"

Line 488: "The R294E mutation, predicted to decrease the stability of the open-interface, decreased twitching motility, pilin accumulation, and phage resistance, consistent with a pilus depolymerization defect. In contrast, the E258A mutation, also predicted to decrease the stability of the open-interface, decreased twitching motility but allowed approximately wild-type levels of pilin accumulation and phage susceptibility, consistent with a pilus polymerization defect, or more plausibly a moderate defect in PilT function that did not cause an obvious overabundance of extracellular pilin."

For the PilC binding, where an effect on PilC binding is observed, can a direct role for these residues in binding PilC be ruled out? It should also be noted that the bacterial 2 hybrid may not necessarily reproduce the in vivo PilT-PilC interaction, where it could be affected by the presence of other pilus biogenesis components. In general, interpretation of these results is difficult because our current knowledge of the structural details of the PilT-PilC interaction and the ATPase reaction cycle is rudimentary.

RESPONSE: Since the structure of PilC bound to PilT is not yet known, no residue can be ruled out as directly interacting with PilC. However, all of the mutations are located between packing units and not plausibly accessible to PilC, which is thought to bind to the central pore of PilT. Thus this study helps to bring clarity to this interaction indicating the conformations of PilT important for binding PilC. The following was added to the text to clarify this issue.

Line 561: "These residues are not in the pore of PilT and thus unlikely to be important for directly contacting PilC, although confirmation of this hypothesis awaits a PilT-PilC co-structure. However, these results are consistent with ●●●●● and ○○○○○-interface-containing conformations being important for binding PilC."

REVIEWER 2

The type IV pili machine has important functions in motility, biofilm formation and virulence. It stimulates the extension and retraction of type IV pili in a large number of bacteria. Generally, these two processes are stimulated by the ATPases PilB for extending a pilus and PilT for retracting a pilus. Over the years, several structures of these and homologous ATPases have been determined. The structures suggest that these ATPases function as hexamers that undergo large conformational changes during ATP binding and hydrolysis. Moreover, it has previously been suggested that PilB and PilT rotate the inner membrane protein PilC in opposite directions to stimulate the incorporation of pilin subunits at the base of the growing pilus or to remove pilin subunit from the base of the shrinking pilus during retractions. However, the details of the interaction to PilC and how PilB/PilT would rotate PilC are not well-understood.

Here, the authors solve the structure of PilT from *Geobacter metallireducens* using crystallography and single particle EM. Different conformations are observed. These structural analyzes were followed up by mutagenesis of residues at identified interfaces using PilT in *P. aeruginosa* and phenotypic characterization. Effects on PilT function is observed as a consequence of these substitutions.

Semantically, I am not sure that referring to the various ATPase discussed as motor PilT-like ATPases is helpful. After all, PilT is the least widespread of the PilB, GspE and PilT ATPases. From sequence alignments, it is also clear that these three ATPases are very different.

RESPONSE: As mentioned above, 'PilT-like ATPases' is the name of this family of homologous ATPases. Thus, we have replaced PilT-like ATPases with PilT/VirB11-like ATPases throughout the text where relevant. While PilT/VirB11 ATPases can be very different, they share homology throughout their motor domains.

For a non-structural biologist the manuscript is incredibly difficult to read and very condense. Also, in the figures it is really difficult to follow and appreciate the differences that the authors are discussing. So (again from a non-structural biologist's point of view), it does not really become clear how the new PilT structures are different from previous structures...

RESPONSE: To help the non-structural biologist follow the subtle distinctions between PilT conformations, we have included block cartoons depicting the indicated conformations in Figures 2, 3, and 6. We also show how they are quantitatively different or similar from previous structures (See Figure 3 and Supplementary Table 1).

...and which new lessons have been learned.

RESPONSE: We have created a new way for researchers to classify the differences in PilT structure easily and reproducibly across this diverse family of ATPases. We showed that the conformations that have been observed crystallographically are not necessarily reproducible in solution. This has major ramifications for interpreting structure-function relationships across the entire family of PilT/VirB11-like ATPases, which have been ascribed various molecular mechanisms based on individual crystal structures. Finally, we use co-evolution analysis and site-mutagenesis to demonstrate the in vivo relevance of specific conformations. These are all new and important lessons learned. Anyone seeking to understand the conformation or molecular mechanism of any PilT/VirB11-like family member will find this information useful. We have added the following text to the discussion to emphasis this point.

Line 586: "By extrapolation, these findings have major ramifications for interpretation of other PiIT/VirB11-like crystal structures, which have individually been used to suggest idiosyncratic molecular mechanisms."

We had also included the following sentence in the discussion.

Line 699: "Given that neither PiIT^{Gm} nor PiIT^{Pa} crystallized in the ○○●○○● conformation, individual PiIT/VirB11-like crystal structures should be interpreted with caution in the absence of accompanying cryoEM analysis."

The in vivo analyzes in *P. aeruginosa* are straight-forward and, as expected, most substitutions at the subunit interface interfere with function. Also here, it would be incredibly helpful if the authors would label the residues in Figure 5 in such a way that it is clear in which conformation they expect the PiIT variants to be locked in.

RESPONSE: Labeled as requested.

REVIEWER 3

Summary:

This article by McCallum et al., provides a detailed structural analysis of PiIT-like ATPases. These motors are essential to the function of type IV pili, which in turn are critical for the virulence of many bacterial pathogens. The authors contribute four new crystal structures and provide a thorough cryo-EM analysis leading to four further maps and models of PiIT/PilB ATPases. The authors tie together the structural work presented in this manuscript with previous work and propose a comprehensive model of PiIT function. Furthermore, coevolution analysis and PiIT mutants provide functional validation for the conclusions drawn. The manuscript is clear and well written and provides novel insights to the current state of knowledge of PiIT ATPases. Experiments are well executed and appropriate statistical analyses are provided. This work is a valuable addition to the field and I, therefore, support publication with some minor modifications.

Minor Points:

1. Typo in legend to Figure 1: In the last line, the brackets around I in "I | σ (I)" are missing (Page 4, line 131).

RESPONSE: Fixed as requested.

2. Question, Page 10: the particle distribution ratio between C3OCOCOC and C2OOCOCOC between PiIT^{Gm} and PiIT^{Pa} is significantly different in the cryo-EM analysis. Do the authors have a suggestion as to the functional implication of this observation? Or do the authors believe that the C3OCOCOC conformation is not relevant in vivo at all?

RESPONSE: We are not sure why the particle distribution ratio is different between species. It might be as simple as K58 in PiIT^{Pa} being a glutamine in PiIT^{Gm}, which would fit with the altered distribution in the PiIT^{Pa} K58A mutant. This possibility has been added to the text.

Line 547: “That a single point mutation can shift the ratio between the ○○●○○● and ○●○○○●● conformations is consistent with different ratios of those conformations in PilT^{Gm} and PilT^{Pa}; K58 is a glutamine in PilT^{Gm}, for example.”

3. Page 10, line 410: The authors state that no density consistent with nucleotide was present in the maps and that this is presumably due to the low resolution of the maps (4.0 and 4.1 Å). However in the 4.4 Å structure of PilT^{Gm} briefly incubated with ATP, the nucleotide density is visible. Therefore, I would be more inclined to attribute the lack of nucleotide density in the PilT^{Gm} without nucleotide structure to the fact that no nucleotide was added to begin with and/or that nucleotide that was potentially carried over from the E. coli expression system is present at low occupancy. Perhaps this sentence can be modified accordingly.

RESPONSE: This sentence has been modified accordingly.

4. Page 20, cryo-EM analysis section: Was the microscope equipped with an energy filter? If so, what was the slit width? What was the total electron dose in e/Å² ?

RESPONSE: The microscope was not equipped with an energy filter. The exposure (i.e. total electron dose) is indicated in Supplementary Table 4 in e/Å².

5. Page 20, line 873: Perhaps reword defocuses to defoci or defocus range or values.

RESPONSE: Modified ‘defocuses’ to ‘defocus values’.

6. Page 21, line 914: typo. “...adherent bacteria were stained...”

RESPONSE: This sentence has been modified accordingly.

7. Page 27, line 993: typo? Should it read “...unit is composed of the N2Dn and CTDn+1...”

RESPONSE: This sentence has been modified accordingly.

REVIEWERS' COMMENTS:

Reviewer #1 (Remarks to the Author):

The authors have responded clearly to the points which I raised previously. I have no remaining concerns about the methodology employed or conclusions drawn.
Jeremy Derrick